# Reducing Task Discrepancy of Text Encoders for Zero-Shot Composed Image Retrieval

## Abstract

Composed Image Retrieval (CIR) aims to retrieve a target image based on a reference image and conditioning text, enabling controllable image searches. Due to the expensive dataset construction cost for CIR triplets, a zero-shot (ZS) CIR setting has been actively studied to eliminate the need for human-collected triplet training datasets of the target domain. The mainstream methods of ZS-CIR research typically employ a projection module that projects a CLIP image embedding to the CLIP text token embedding space while all encoders are fixed. Using such a projected embedding, those methods then generate an image-text composed feature, which is used as a query for retrieval. However, we point out that using fixed CLIP encoders for ZS-CIR has an inherent limitation since there exists a significant task discrepancy between the original pre-training task of the encoders (text $\leftrightarrow$ image) and the target CIR task (image + text $\leftrightarrow$ image). To reduce such a discrepancy, a naive solution would be to train both image and text encoders with CIR triplets in a supervised manner. Instead, we introduce the Reducing Task Discrepancy of text encoders for Zero-Shot Composed Image Retrieval (**RTD**), an efficient post-precessing approach designed to enhance the capability of text encoders for ZS-CIR. Namely, we devise a novel target-anchored text contrastive learning, which solely updates the text encoder using cheap *text* triplets, consisting of reference and target texts instead of images. We also introduce two enhancements to this approach: a refined batch sampling strategy and a sophisticated concatenation scheme. Integrating RTD into existing projection-based ZS-CIR methods significantly improves performance across various datasets and backbones, achieving competitive or superior results compared to other resource-intensive state-of-the-art CIR methods beyond projection-based approaches.

## 1 Introduction

Composed Image Retrieval (CIR) is an emerging task aimed at retrieving a target image that closely resembles a reference image while reflecting the changes described in a conditioning text. Using a query composed of image and text allows users to conduct more precise and flexible searches by specifying the desired modifications to the image through text. Supervised CIR methods (Baldrati et al., 2022a; Delmas et al., 2022; Lee et al., 2021) have been introduced to fuse the information from the bi-modal query, using labeled data from the target domain in the form of triplets $(I_r, T_c, I_t)$, in which $I_r$ is a reference image, $T_c$ is a conditioning text, and $I_t$ is a target image. However, unlike thr typical web-crawled image-text datasets (Schuhmann et al., 2022b), acquiring sufficient triplets for training needs expensive manual human annotations. Hence, the existing CIR triplet datasets are typically small, limiting the cability of supervised approaches trained on such datasets.

To overcome the dependency on small-scale, human-verified triplets of the target domain, a new task, Zero-Shot Composed Image Retrieval (ZS-CIR), has been recently introduced. The first approach for this task utilizes the power of recent vision-language (VL) generative models. For example, a line of studies (Gu et al., 2023; Ventura et al., 2024; Levy et al., 2024; Zhang et al., 2024) uses text-to-image models like IP2P (Brooks et al., 2023) to synthesize large-scale CIR triplets for training, in place of the target CIR triplet datasets. Another example can be found in CIReVL (Karthik et al., 2023), which eliminates the need for training by using image captioning models and large-language models (LLM) during inference. While these methods achieve decent performance, they are impractical due to high computational and memory requirements for utilizing generative models.

Figure 1: The task discrepancy of projection-based ZS-CIR methods between the pre-training task (image-text alignment) and the ZS-CIR task (image-text composition).

The second approach for removing the dependency on the CIR triplet datasets, which has become the mainstream due to its simplicity, employs an integrable projection module on top of the pre-trained, frozen, and shared VL embedding space, such as CLIP (Radford et al., 2021). Namely, a projection module $\phi$, which maps a CLIP image embedding to the CLIP text token embedding space, can be trained by solely using images (Saito et al., 2023; Baldrati et al., 2023) or texts (Gu et al., 2024). During inference, as illustrated in Figure 1, these methods first project the embedding of the query image to a text token embedding [\$] using the function $\phi$. This embedding is then combined with the conditioning text $[T_c]$ to create the prompt "a photo of [\$] that $[T_c]$", which is used as a query for the text-to-image retrieval.

The core assumption of the above second approach, which often is referred to as projection-based ZS-CIR (Saito et al., 2023; Baldrati et al., 2023; Gu et al., 2024), is that the pre-trained text encoder should be robust enough to combine information from both the projected text token embedding and the conditioning text. However, we argue that this can cause significant *task discrepancy* for the pre-trained text encoder between the original image-text alignment pre-training task (of CLIP) and the ZS-CIR task. For example, in Figure 1, consider an *ideal caption* that accurately describes the target image. Since the CLIP text and image encoders are learned through contrastive learning, we can expect that the target image embedding (Fig. 1c) will align well with the embedding (Fig. 1b) of the ideal caption. In contrast, in the projection-based ZS-CIR, the text encoder receives a concatenated caption that combines the projected token [\$] and the conditioning text, in place of the ideal caption. However, the text encoder typically is not trained to encode complex textual modifications—such as addition, negation, or spatial relationships—to the reference image, which are common in conditioning texts. As a result, there is no guarantee that the textual embedding of the concatenated caption (Fig. 1a) closely aligns with that of the target image embedding (Fig. 1c), which will undermine the final retrieval performance.

To that end, we propose a post-processing approach that can be directly applied to existing projection-based ZS-CIR methods, reducing the task discrepancy of the text encoder only with cheap *text* triplets. These triplets $(T_r, T_c, T_t)$ – in which $T_r$, $T_c$ and $T_t$ is a reference caption, a conditioning text, and a target caption, respectively – can be automatically generated without human labor (Liu et al., 2021; Wu et al., 2021) and intensive resources (Gu et al., 2023; Ventura et al., 2024; Levy et al., 2024; Zhang et al., 2024), but with simple rule-based templates and reference captions $T_r$. Using these triplets, we introduce a *target-anchored text contrastive learning*, which trains the text encoder to update the embeddings of the concatenated caption $T_{r+c}$ (formed by concatenation of reference caption $T_r$ and conditioning text $T_c$) to align closely with the fixed embedding of the target caption $T_t$, which serves as an anchor point obtained from the frozen text encoder. We also propose two techniques to enhance the effectiveness of such language-only supervision further: a batch sampling strategy that incorporates hard negatives in each mini-batch and a refined concatenation scheme for $T_r$ and $T_c$ to improve generalization capability. We note our approach can be seamlessly integrated with existing projection-based ZS-CIR methods (Saito et al., 2023; Baldrati et al., 2023; Gu et al., 2024) by replacing their text encoder with our updated text encoder while fixing other modules, *e.g.*, the image encoder and $\phi$. Moreover, our approach is highly efficient in the training process due to the benefits of language-only training, as highlighted by (Gu et al., 2024).

Our experimental results demonstrate that our proposed method, dubbed as **RTD** (Reducing Task Discrepancy of text encoders for Zero-Shot Composed Image Retrieval), substantially improves the ZS-CIR performance in diverse evaluation datasets (CIRR (Liu et al., 2021), CIRCO (Baldrati et al., 2023), FashionIQ (Wu et al., 2021), COCO object composition (Saito et al., 2023), and GeneCIS (Vaze et al., 2023)). Namely, when integrated into the existing projection-based ZS-CIR methods (SEARLE (Baldrati et al., 2023), Pic2Word (Saito et al., 2023), and LinCIR (Gu et al., 2024)), RTD consistently enhances performance across different size backbones, underscoring the generality of our approach. Compared to other CIR methods beyond projection-based ZS-CIR approaches, particularly those based on synthetic training CIR triplets, RTD delivers comparable or superior performance with significantly higher efficiency. Our systematic ablation analyses reveal that the performance enhancement primarily results from reducing the task discrepancy of the text encoder, rather than merely tuning the textual backbone network with additional data. We investigate the impact of various text triplet generation strategies and verify that RTD consistently improves ZS-CIR performances across them. Moreover, instead of updating all parameters of the text encoder, we show that a more efficient approach, which selectively updates only a few layers of the text encoder, can be effective as well.

## 2 RELATED WORK

**Projection-based ZS-CIR methods.** Pic2Word (Saito et al., 2023), SEARLE (Baldrati et al., 2023), and LinCIR (Gu et al., 2024), are built upon the frozen CLIP model, where a projection module $\phi$ is trained without CIR triplets. Each projection-based CIR methods employ a different training scheme for $\phi$ (See Section 4.1 for details). While these approaches demonstrate promising ZS-CIR performances, they are dependent on the pre-trained CLIP visual and text encoders, leading to a task discrepancy between the CLIP pretext task and the CIR task. In this paper, to address this, we devise an efficient text encoder-only updating scheme that utilizes *cheap* text triplets. While our method is built upon projection-based ZS-CIR methods, our approach is also related to another category of ZS-CIR methods (Gu et al., 2023; Ventura et al., 2024; Levy et al., 2024), using synthetically generated CIR triplets instead of the target CIR triplets, with some updating CLIP backbones. However, our method stands out by utilizing text triplets and updating only the text encoder, resulting in significantly more efficient training. Despite its efficiency, our method achieves performance that is competitive with, or even superior to, these other approaches.

**Task discrepancy between the CLIP pretext task and CIR.** Combiner (Baldrati et al., 2022b) updates the text encoder to minimize the gap between the target caption feature and the summation of the reference image feature and the instruction text feature. However, Combiner needs expensive CIR triplets $(I_r, T_c, I_t)$ for training. Our approach uses text-only triplets $(T_r, T_c, T_t)$, cheap and automatically generated. As another example, Chen & Lai (2023) synthesizes a triplet of an original image, the corresponding caption, and the masked image, where treating the original image as the target image, the caption as the conditioning text, and the masked image as the reference image. This approach, however, still has a gap between conditioning text (*e.g.*, "change the dog to a cat") and image caption (*e.g.*, "a dog is jumping to catch a frisbee"); furthermore, it needs the full fine-tuning of the CLIP model, resulting in changing the visual embeddings in the retrieval database. On the other hand, RTD directly uses the instruction texts for training and does not change the target visual encoder, which enables the reuse of pre-extracted CLIP visual embeddings. Lastly, CIReVL (Karthik et al., 2023) reduces the task discrepancy by making a descriptive caption of the composed query using a large captioning model and LLM. Although CIReVL shows great performance without any training, this method needs inefficient and expensive inferences of BLIP (Li et al., 2023) and GPT (Brown et al., 2020). Furthermore, it needs a well-tuned task-specific prompt by a skilled user. RTD is much more efficient than CIReVL and fully automated without direct human intervention.

## 3 MAIN METHOD

### 3.1 OBTAINING TEXT TRIPLETS

To address the task discrepancy described in the Introduction and Figure 1, we aim to employ text triplets $(T_r, T_c, T_t)$, which can be cheaply and automatically generated, instead of directly using

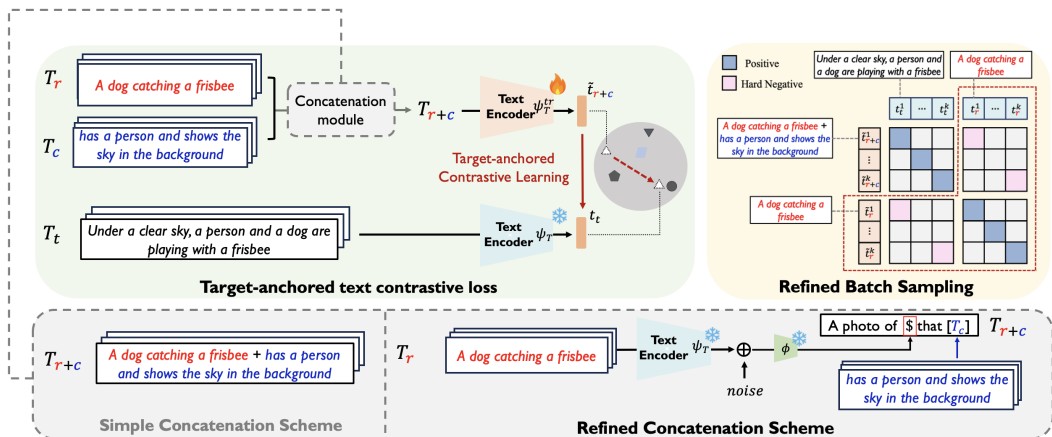

Figure 2: Overview of RTD.

the expensive CIR triplets $(I_r, T_c, I_t)$. There are two strategies to generate these text triplets: via large language models (LLMs) (Gu et al., 2023; Brooks et al., 2023; Levy et al., 2024; Ventura et al., 2024) or, more efficiently, through rule-based templates (Gu et al., 2023). We investigate both generation strategies and demonstrate that RTD consistently improves performance across them. Below, we briefly introduce both strategies, with detailed explanations and examples provided in Appendix A.2, and a comprehensive comparison of generation costs is provided in Appendix C.

For the LLM-generated text triplets, we use the public text triplets proposed by Compodiff (Gu et al., 2023), which are employed in our main experiments unless otherwise noted. These triplets are generated by taking a caption $T_r$ as an input of the fine-tuned LLM, whose output predicts the corresponding conditioning text $T_c$ and the target caption $T_t$. Other works, such as IP2P (Brooks et al., 2023), CoVR (Ventura et al., 2024), and CASE (Levy et al., 2024), have also explored generating text triplets using LLMs, differing in LLM model types, input data, and fine-tuning strategies. The original purpose of these text triplet generation is to construct CIR triplets $(I_r, T_c, I_t)$, but they also release their text triplets used for their CIR triplet construction. In addition to these publicly available text triplets, we also implement and evaluate an efficient in-context learning-based generation strategy using LLaMA3-8B (Dubey et al., 2024), which removes the need for a fine-tuning phase. We conduct experiments with all the aforementioned text triplets in Table 7 and observe that RTD consistently delivers significant performance enhancements, demonstrating the reproducibility and effectiveness of our approach.

We also investigate LLM-less text triplet generation strategies. For example, we can extract a "keyword" (*e.g.*, nouns) from a given caption and randomly change the keyword from a randomly chosen keyword (Gu et al., 2023). The modification caption is automatically generated by using predefined templates (*e.g.*, "change [original keyword] to [altered keyword]"). Our experiments show that this simple rule-based approach performs similarly to the LLM-based approach.

### 3.2 TARGET-ANCHORED TEXT CONTRASTIVE LEARNING

Now, we explain our approach to update the text encoder for mitigating the task discrepancy solely with the generated text triplets $(T_r, T_c, T_t)$. We first assume that there exists a pre-trained projection module $\phi$ obtained by the projection-based ZS-CIR methods (Saito et al., 2023; Baldrati et al., 2023; Gu et al., 2024). Recall that for a given reference image $I_r$ and conditioning text $T_c$, the final composed feature is generated by passing the text prompt "a photo of $\phi(\psi_V(I_r))$ that $T_c$" to the text encoder $\psi_T$, where $\psi_V$ is the visual encoder and $\phi$ is the projection module (See Figure 1). We aim to update the text encoder $\psi_T$ to reduce the discrepancy between the pretext task and ZS-CIR task using the text triplets while maintaining $\psi_V$ and $\phi$ frozen.

**[Target-anchored text contrastive loss]** We apply contrastive learning using a paired caption $(T_{r+c}, T_t)$, where $T_{r+c}$ denotes a concatenated caption of reference caption $T_r$ and conditioning caption $T_c$. Namely, we let the representation of the concatenated caption closely approximate that of the target caption. However, solely updating the text encoder while fixing the image encoder can break the alignment between image and text encoders. To prevent the issue, we extract the text

embedding of $T_t$ using the frozen text encoder $\psi_T$, while the concatenated caption $T_{r+c}$ is extracted from the learnable text encoder $\psi_T^{tr}$, initialized from $\psi_T$. Here, we assume that as the target caption $T_t$ is a standard caption, a text embedding $\psi_T(T_t)$, is well-aligned with the frozen image embedding space. Following the assumption, we fix the target textual embedding to serve as an anchor point. This approach helps maintain the pre-trained alignment while learning new relationships. As shown in Section 4.4, this anchoring is essential for fine-tuning the text encoder with our objective.

Now, we introduce our target-anchored text contrastive loss $\mathcal{L}_{TCL}$ using two text encoders: a frozen pre-trained text encoder $\psi_T$ and a learnable text encoder $\psi_T^{tr}$ which is initialized with $\psi_T$. Textual latent embeddings $\tilde{t}_{r+c}$ and $t_t$ are extracted from $\psi_T^{tr}$ and $\psi_T$, respectively. Namely, $\tilde{t}_{r+c} = \psi_T^{tr}(E_w^{tr}(T_{r+c}))$ and $t_t = \psi_T(E_w(T_t))$, where $E_w$ is a word embedding layer. We aim to tune $\psi_T^{tr}$ to minimize the distance between the concatenated textual embedding $\tilde{t}_{r+c}$ and the target textual embedding $t_t$ while maximizing the distance from other textual embeddings within the batch. We employ a symmetric InfoNCE loss (Chen et al., 2020; Cohen et al., 2022), as follows:

$$
\mathcal{L}_{TCL} = \frac{1}{B} \sum_{k=1}^{B} - \log \frac{e^{(c(\tilde{t}_{r+c}^k, t_t^k)/\tau)}}{\sum_{j=1}^{B} e^{(c(\tilde{t}_{r+c}^k, t_t^j)/\tau)} + \sum_{j \neq k} e^{(c(t_t^k, t_t^j)/\tau)}} - \log \frac{e^{(c(t_t^k, \tilde{t}_{r+c}^k)/\tau)}}{\sum_{j=1}^{B} e^{(c(t_t^k, \tilde{t}_{r+c}^j)/\tau)} + \sum_{j \neq k} e^{(c(\tilde{t}_{r+c}^k, \tilde{t}_{r+c}^j)/\tau)}}
$$

(1)

where $c(\cdot, \cdot)$ denotes the cosine similarity, $B$ is the batch size, and $\tau$ is a temperature.

**[Refined batch sampling strategy for hard negatives]** To further enhance the efficacy of updating the text encoder, we devise a simple yet effective batch sampling strategy that incorporates pairs of $(T_{r+c}, T_t)$ and $(T_r, T_r)$ within the same batch. For example, as presented in Figure 2, a pair such as $(T_{r+c}$: "A dog catching a frisbee + has a person and shows in the background", $T_t$: "Under a clear sky, a person and a dog are playing with a dog") is sampled along with its corresponding reference pair $(T_r$: "A dog catching a frisbee", $T_r$: "A dog catching a frisbee") in the same batch. This setup ensures that the concatenated text $T_{r+c}$ and its corresponding reference text $T_r$ implicitly act as hard negatives for each other, as their semantics are much more similar ($T_{r+c}$ is derived from $T_r$) than those of other randomly sampled texts in the batch. Moreover, we believe including $(T_r, T_r)$ pairs in the contrastive learning helps the learnable text encoder $\psi_T^{tr}$ remain closely aligned with the pre-trained encoder $\psi_T$.

**[Refined concatenation of reference and conditioning texts]** A naive concatenation strategy also can suffer from training-inference task discrepancy because we actually use "a photo of [$] that [$T_c$]" for inference. To tackle this issue, rather than simply concatenating the $T_r$ and $T_c$, we also use the prompt "a photo of [$] that [$T_c$]" for updating the text encoder, where [$] is obtained by the reference caption $T_r$ with the projection module $\phi$. Instead of obtaining a pseudo-word token with latent image embedding $v$, we utilize a textual latent embedding from the reference caption $T_r$, *i.e.*, $\phi(t_r)$. However, Gu et al. (2024) showed that naively replacing the image encoder with the text encoder for the input of $\phi$ will suffer from the modality gap (Liang et al., 2022), a phenomenon where text and image embeddings have a gap between them. Thus, to reduce the potential negative effect of the modality gap, following Gu et al. (2024), we inject random noise into the textual token representation before it is processed by $\phi$. More analyses on variations of noise are in Appendix B.6.

Figure 2 illustrates the overview of RTD. We use CLIP backbone and pre-trained projection module $\phi$ produced by the existing projection-based ZS-CIR methods. The text encoder is trained using the proposed loss function (Eq. (1)) while applying the refined batch sampling and concatenation scheme. During inference, the procedure mirrors that of existing ZS-CIR methods, except we utilize the updated text encoder $\psi_T^{tr}$ instead of the frozen one $\psi_T$. Note that our method only updates the text encoder while the image encoder and the projection module are frozen.

*Remark.* We note that our entire training process is highly efficient due to the advantages of language-only training as highlighted by Gu et al. (2024). First, the generation cost of text triplets we use is significantly lower than that of CIR triplets. Namely, text triplet generation avoids resource-intensive text-to-image generation (Gu et al., 2023; Brooks et al., 2023), making it 15 times faster than CIR triplet generation, in the process used in CompoDiff (Gu et al., 2023). If we opt for the rule-based text triplet generation approach, efficiency is further enhanced by eliminating the LLM fine-tuning and generation steps, making the process 570 times faster—generating 1M text

triplets takes just 0.1 hours—compared to the CIR triplet generation case. Moreover, the text triplets we used take up only 100MB, whereas storing a similar quantity of images requires significantly more space (*e.g.*, around 400GB in the case of CC3M (Sharma et al., 2018)). Second, the training complexity for the text encoder is substantially lower than that for the visual encoder due to the relatively short token lengths of texts ($\sim$12) compared to images (256). The average inference time of the CLIP ViT-L/14 image encoder is $\times$ 3.5 times slower than that of the text encoder. As a result, the additional training cost of RTD is small: 0.5 hours using 8 A100 for CLIP ViT-L/14, which is reasonable compared to the original training times of projection-based ZS-CIR methods: LinCIR (0.5 hours), SEARLE (4.3 hours), and Pic2Word (3 hours). Further details and analyses on training efficiency are provided in Appendix C. Moreover, in Appendix B.3, we present a more efficient implementation option by selectively updating only a few layers of the text encoder.

### 3.3 Can RTD really reduce the task discrepancy of the text encoder?

In this subsection, we quantitatively verify whether RTD indeed reduces the task discrepancy. We first conduct a toy experiment that measures the text-to-image (T2I) retrieval performance of the text encoder with conditional texts. We retrieve the target images $I_t$ with the concatenated text query $T_{r+c}$ or the ideal target caption $T_t$. If our text encoder successfully

Table 1: T2I retrieval performance of different text encoders on CIRCO validation dataset.

| Query | Text encoder | mAP@5 | mAP@10 | mAP@25 |
|-------|--------------|-------|--------|--------|
| $T_t$ | Frozen | 18.96 | 19.31 | 21.05 |
| $T_{r+c}$ | Frozen | 10.12 | 10.71 | 12.34 |
| $T_{r+c}$ | RTD | 15.12 | 15.80 | 17.77 |

handles the discrepancy due to the concatenated caption, the text encoder updated by RTD will perform better than the frozen one. We use the CLIP ViT-L/14 and CIRCO (Baldrati et al., 2023) validation dataset for evaluation. Since the CIRCO dataset only has CIR triplets $(I_r, T_c, I_t)$, we use the BLIP (Li et al., 2022) captioner to generate $T_r$ and $T_t$ corresponding to the $I_r$ and $I_t$, respectively. Here, the simple concatenation scheme is applied for the text query $T_{r+c}$ in all cases for a fair comparison. Table 1 shows that when the text encoder is frozen, the retrieval results using the concatenated caption $T_{r+c}$ are significantly worse than those using the target caption $T_t$. It supports the claim that the frozen text encoder suffers from the negative effects of task discrepancy between the pretext task and the CIR task. In contrast, the text encoder updated by RTD shows a significant improvement over the frozen text encoder, showing that it successfully reduces the task discrepancy.

We additionally measure the average cosine similarity between the composed textual features with the prompt "a photo of $\phi(\psi_V(I_r))$ that $T_c$" (Fig. 1a) and the target image features (Fig. 1c). The similarity is measured by the LinCIR ViT-L/14 model on the CIRCO validation split. When we use the frozen CLIP text encoder ($\psi_T$), the average similarity is 0.1. By changing the text encoder to our updated text encoder ($\psi_T^{tr}$), the similarity becomes 0.29 (+0.19). This result shows again that RTD successfully aligns the composed query features using $\phi$ to the frozen CLIP image features.

Lastly, since RTD updates the text encoder backbone (unlike prior projection-based ZS-CIR methods that freeze both backbones), some may question whether the performance gains are due to reducing task discrepancy or simply from updating the text encoder. As will be shown in Table 8, the effectiveness of RTD comes from reducing task discrepancy, not from simple text encoder backbone tuning. We will further elaborate on this point in Section 4.5.

## 4 Experiments

### 4.1 Experimental setup

**Implementation details.** We use the AdamW optimizer (Loshchilov & Hutter, 2019) with a weight decay of 0.01. The learning rate is set to $10^{-5}$, with a batch size of 512. For a fair comparison, we select the text encoder model with the best zero-shot CIRR (Liu et al., 2021) dev R@1 score for evaluating RTD. We evaluate the CIR performances of the model in a zero-shot manner by evaluating it across five different benchmarks. We mainly use the visual and textual encoders of the CLIP ViT-B/32 and ViT-L/14 (Radford et al., 2021) as our backbone. Unless otherwise noted, we use LLM-based 2.5M text triplets provided by CompoDiff (Gu et al., 2023) for the training. We set the $\tau$ as 0.07 in Eq. (1) and scale the standard deviation of Gaussian distribution as 0.5 for the noise

Table 2: **FashionIQ validation results.** The results of RTD combined with Pic2Word (Saito et al., 2023), SEARLE (Baldrati et al., 2023), and LinCIR (Gu et al., 2024) across different CLIP backbones (ViT-B/32 and ViT-L/14) are shown. Blue denotes the performance gain achieved by RTD.

| | | Shirt | | Dress | | Toptee | | Average | |
|---|---|---|---|---|---|---|---|---|---|
| | | R@10 | R@50 | R@10 | R@50 | R@10 | R@50 | R@10 | R@50 |
| ViT-B/32 | Pic2Word | 13.40 | 28.46 | 8.48 | 20.77 | 13.31 | 29.68 | 11.73 | 26.30 |
| | +RTD | 23.06 (+9.66) | 40.48 (+12.02) | 20.33 (+11.85) | 41.75 (+20.98) | 24.12 (+10.81) | 46.35 (+16.67) | 22.5 (+10.77) | 42.86 (+16.56) |
| | SEARLE | 24.78 | 41.85 | 17.90 | 36.99 | 25.24 | 46.71 | 22.64 | 41.85 |
| | +RTD | 26.69 (+1.91) | 44.31 (+2.46) | 20.72 (+2.82) | 43.13 (+6.14) | 26.67 (+1.43) | 48.75 (+2.04) | 24.7 (+2.06) | 45.4 (+3.55) |
| | LinCIR | 18.55 | 34.64 | 15.67 | 33.86 | 20.19 | 40.08 | 18.14 | 36.20 |
| | +RTD | 23.65 (+5.10) | 42.74 (+8.10) | 19.98 (+4.31) | 41.75 (+7.89) | 24.73 (+4.54) | 46.56 (+6.48) | 22.79 (+4.65) | 43.68 (+7.48) |
| ViT-L/14 | Pic2Word | 26.59 | 42.93 | 21.32 | 43.53 | 28.10 | 48.19 | 25.34 | 44.88 |
| | +RTD | 27.97 (+1.38) | 46.96 (+4.03) | 23.50 (+2.18) | 46.65 (+3.12) | 31.31 (+3.21) | 53.09 (+4.90) | 27.59 (+2.25) | 48.90 (+4.02) |
| | SEARLE | 26.94 | 45.34 | 19.58 | 40.80 | 28.45 | 49.77 | 24.99 | 45.30 |
| | +RTD | 32.63 (+5.69) | 50.39 (+5.05) | 23.2 (+3.62) | 47.25 (+6.45) | 32.18 (+3.73) | 54.56 (+4.79) | 29.34 (+4.35) | 50.73 (+5.43) |
| | LinCIR | 30.42 | 47.99 | 21.86 | 44.77 | 29.98 | 50.38 | 27.42 | 47.71 |
| | +RTD | 32.83 (+2.41) | 50.44 (+2.45) | 24.49 (+2.63) | 48.24 (+3.47) | 33.4 (+3.42) | 54.56 (+4.18) | 30.24 (+2.82) | 51.08 (+3.37) |

injection. More results on various noise distributions can be found in the Appendix. All experiments were conducted using four NVIDIA A100 GPUs with Python 3.8 and Pytorch (Paszke et al., 2019).

**Evaluation datasets and metrics.** We compare ZS-CIR methods on five benchmark datasets: CIRR (Liu et al., 2021), CIRCO (Baldrati et al., 2023), FashionIQ (Wu et al., 2021), COCO object composition (Saito et al., 2023), and GeneCIS (Vaze et al., 2023). Details of each dataset are in the Appendix A.1. For CIRR, FashionIQ, COCO, and GeneCIS, we have reported their recall scores at the top K retrieval results (R@K). Since the CIRCO dataset includes multiple positive images for each query, we use a ranking-based metric—mean Average Precision scores at the top K results (mAP@K)—which provides a more robust and reliable assessment (Musgrave et al., 2020; Chun et al., 2022). For the main results, we compare the results on the three categories (Shirt, Dress, Toptee) of the FashionIQ validation split, as well as the test sets of CIRR and CIRCO. For the ablation studies and analyses, the validation splits of these three datasets are utilized. GeneCIS and COCO object composition results and their detailed explanations can be found in the Appendix B.1.

**Baselines.** We evaluate the effect of our method when combined with three representative projection-based ZS-CIR methods: Pic2Word (Saito et al., 2023), SEARLE (Baldrati et al., 2023), and LinCIR (Gu et al., 2024). All three methods share the same core concept shown in Figure 1, but use different training schemes. Pic2Word(Saito et al., 2023) optimizes contrastive loss between the image embedding and its projected text embedding of "a photo of [$]" to obtain the projection module $\phi$. Similarly, SEARLE (Baldrati et al., 2023) employs a two-stage approach, starting with an optimization-based textual inversion phase followed by a distillation phase for the projection module $\phi$. LinCIR (Gu et al., 2024) introduces a language-only self-supervised task involving keyword token replacement by letting the original text embedding and the replaced text embedding whose keyword tokens are changed to the projected original text embedding by $\phi$.

We train all these methods with the same backbone (CLIP ViT-B/32 and ViT-L/14). For LinCIR, we also conduct experiments with a larger backbone (ViT-G/14), enabled by its fast training capability. We use the publicly available pre-trained model for SEARLE (ViT-B/32, ViT-L/14) and Pic2Word (ViT-L/14). Otherwise, we reproduce the results using the official implementation. When reproducing, we adhere to the same settings in the original papers. For example, we select the final last epoch model for the Pic2Word ViT-B/32 model and choose the model based on the best zero-shot CIRR dev R@1 score for LinCIR. Moreover, we compare our method with a broader range of CIR methods; (1) recent projection-based ZS-CIR methods: KEDs (Suo et al., 2024), Context-I2W (Tang et al., 2024), (2) attempt to tackle task discrepancy: MTCIR (Chen & Lai, 2023), (3) synthetically generated CIR triplets-based approach: CoVR (Ventura et al., 2024), CASE (Levy et al., 2024), Compodiff (Gu et al., 2023), and (4) training-free approach: CIReVL (Karthik et al., 2023).

## 4.2 MAIN RESULTS: INTEGRATION WITH ZS-CIR METHODS

Table 2 summarizes the evaluation results on the FashionIQ dataset. In the table, we observe that the incorporation of our approach with ZS-CIR methods significantly improves the performance across all three existing ZS-CIR methods (SEARLE, Pic2Word, and LinCIR) and all backbones (ViT-B/32 and ViT-L/14). For example, regardless of the choice of ZS-CIR methods and backbones, the

Table 3: **CIRR and CIRCO test results.** Details are the same as Table 2.

| | | CIRR | | | CIRCO | | | |
|---|---|---|---|---|---|---|---|---|
| | | R@1 | R@5 | R@10 | mAP@5 | mAP@10 | mAP@25 | mAP@50 |
| ViT-B/32 | Pic2Word | 13.64 | 37.45 | 52.22 | 2.85 | 3.24 | 3.89 | 4.31 |
| | +RTD | 23.59 (+**9.95**) | 51.76 (+**14.31**) | 65.16 (+**12.94**) | 6.39 (+**3.54**) | 6.66 (+**3.42**) | 7.64 (+**3.75**) | 8.16 (+**3.85**) |
| | SEARLE | 23.71 | 53.3 | 66.84 | 8.90 | 9.42 | 10.64 | 11.34 |
| | +RTD | 26.29 (+**2.58**) | 56.41 (+**3.11**) | 69.74 (+**2.90**) | 11.26 (+**2.36**) | 12.11 (+**2.69**) | 13.63 (+**2.99**) | 14.37 (+**3.03**) |
| | LinCIR | 18.87 | 45.66 | 58.43 | 6.25 | 6.74 | 7.62 | 8.10 |
| | +RTD | 24.82 (+**5.95**) | 53.47 (+**7.81**) | 66.87 (+**8.44**) | 8.94 (+**2.69**) | 9.35 (+**2.61**) | 10.57 (+**2.95**) | 11.21 (+**3.11**) |
| ViT-L/14 | Pic2Word | 24.22 | 51.49 | 64.05 | 8.27 | 9.10 | 10.09 | 10.75 |
| | +RTD | 27.86 (+**3.64**) | 56.24 (+**4.75**) | 68.48 (+**4.43**) | 9.13 (+**0.86**) | 9.63 (+**0.53**) | 10.68 (+**0.59**) | 11.27 (+**0.52**) |
| | SEARLE | 24.89 | 52.31 | 65.69 | 11.62 | 12.72 | 14.33 | 15.13 |
| | +RTD | 26.63 (+**1.74**) | 56.17 (+**3.86**) | 68.96 (+**3.27**) | 16.53 (+**4.91**) | 17.89 (+**5.17**) | 19.77 (+**5.44**) | 20.68 (+**5.55**) |
| | LinCIR | 23.76 | 52.89 | 66.46 | 13.00 | 14.11 | 15.81 | 16.68 |
| | +RTD | 26.63 (+**2.87**) | 56.17 (+**3.28**) | 68.96 (+**2.50**) | 17.11 (+**4.11**) | 18.11 (+**4.00**) | 20.06 (+**4.25**) | 21.01 (+**4.33**) |

Table 4: **Results on larger backbone.** LinCIR with OpenCLIP ViT-G/14 (Ilharco et al., 2021). We use the FashionIQ validation split, as well as the test splits of CIRR and CIRCO, for evaluation.

| Method | CIRR | | CIRCO | | FashionIQ | | Avg |
|---|---|---|---|---|---|---|---|
| | R@5 | R@10 | mAP@10 | mAP@25 | R@10 | R@50 | |
| LinCIR | 64.51 | 76.12 | 21.93 | 24.12 | 44.53 | 65.53 | 49.46 |
| +RTD | **67.47** (+**2.96**) | **78.31** (+**2.19**) | **22.29** (+**0.36**) | **24.46** (+**0.34**) | **46.21** (+**1.68**) | **67.26** (+**1.73**) | **50.99** (+**1.53**) |

minimum performance gain for average R@10 and R@50 scores is greater than 2 and 3.5 points, respectively. Table 3 shows a similar trend on the CIRR and CIRCO datasets. Notably, in some metrics on the CIRR and CIRCO datasets, the performance improvements achieved through our method (ViT-B/32) even exceed those obtained by employing a larger backbone (ViT-L/14), which demonstrates the effect of our method. Specifically, in the CIRR R@1 score, SEARLE + RTD (26.29) and LinCIR + RTD (24.82) using ViT-B/32 surpasses the original results of SEARLE (24.89) and LinCIR (23.76) using ViT-L/14. We verify that a similar tendency is observed in the GeneCIS and COCO object composition task datasets, as detailed in the Appendix.

We further evaluate the performance of RTD using the significantly larger backbone (ViT-G/14). In Table 4, combining RTD and LinCIR (chosen due to its fast training capabilities) achieves strong ZS-CIR performances on all benchmarks. Details and the full results are provided in the Appendix B.2. We also provide additional qualitative retrieval results in the Appendix D.

## 4.3 MAIN RESULTS: COMPARISON WITH STATE-OF-THE-ARTS

Table 5 shows the overview of comparison results with state-of-the-art CIR methods. First, we observe that LinCIR + RTD outperforms recent projection-based ZS-CIR methods (KEDs and Context-I2W), which use external knowledge databases or complex projection modules. Additionally, RTD even can also be integrated with them. We conduct experiments only with Context-I2W since KEDs do not provide pre-trained weights, and observe that Context-I2W + RTD considerably enhances the performance of Context-I2W, further demonstrating the compatibility of RTD. Second, RTD offers competitive performance while being more computationally efficient compared to MT-CIR, which shares a similar motivation, and methods that rely on synthetically generated CIR triplets (CoVR, CASE, and Compodiff). As noted in the Remark section (Section 3), using images during training incurs significant overhead due to the slow inference time of the image encoder (slower than the text encoder). Note that methods like MTCIR and those utilizing synthetic CIR triplets require the forwarding passes of the image encoder during training. Moreover, as detailed in Appendix C, the cost of generating synthetic CIR triplets is notably high, while MT-CIR and CASE necessitate additional updating of the visual features in the retrieval database. In contrast, RTD requires only *cheap* text triplets, without the need for retrieval updates or forwarding passes of the image encoder during training. Third, LinCIR + RTD outperforms training-free CIReVL, whose inference time is 79 times slower and 13 times higher memory usage than existing projection-based methods (including RTD) due to the use of costly inferences from image captioners and LLMs. Note that the incorporation of RTD does not increase inference time because there is no change in the model architecture. Lastly, LinCIR + RTD (CLIP ViT-G) ranks high in every benchmark compared to all other methods. We believe the scalability of RTD, enabling the use of a larger backbone (ViT-G/14), further highlights the simplicity and efficiency, as it also benefits from the language-only training approach highlighted

Table 5: **Comparison with other baselines.** In the "Training data type" column, "$I$ and $T$" denote conventional pair-based images and their corresponding captions (not triplets). Note that this comparison is not entirely fair due to differences in backbone models and training data across categories. The same evaluation datasets are used as in Table 4. Best scores are highlighted in red.

| Category | Method | Arch | Training data type | CIRR | | CIRCO | | FashionIQ | |
|---|---|---|---|---|---|---|---|---|---|
| | | | | R@5 | R@10 | mAP@10 | mAP@25 | R@10 | R@50 |
| | KEDs (Suo et al., 2024) | CLIP ViT-L | $I, T$ | 54.8 | 67.2 | - | - | 26.8 | 47.9 |
| (1) | Context-I2W (Tang et al., 2024) | CLIP ViT-L | $I$ | 55.4 | 68.6 | - | - | 27.9 | 49.1 |
| | **Context-I2W + RTD** | **CLIP ViT-L** | $\langle T_r, T_c, T_t \rangle$ | **58.4** | **70.5** | **-** | **-** | **28.1** | **49.5** |
| | LinCIR | CLIP ViT-L | $T$ | 52.9 | 66.5 | 14.1 | 15.8 | 27.4 | 47.7 |
| | **LinCIR + RTD** | **CLIP ViT-L** | $\langle T_r, T_c, T_t \rangle$ | **56.2** | **69.0** | **18.1** | **20.1** | **30.2** | **51.1** |
| | LinCIR | CLIP ViT-G | $T$ | 64.5 | 76.1 | 21.9 | 24.1 | 44.5 | 65.5 |
| | **LinCIR + RTD** | **CLIP ViT-G** | $\langle T_r, T_c, T_t \rangle$ | **67.5** | **78.3** | 22.3 | 24.5 | 46.2 | 67.3 |
| (2) | MT-CIR (Chen & Lai, 2023) | CLIP ViT-L | $I, T$ | 54.6 | 67.6 | 11.6 | 13.0 | 35.4 | 57.4 |
| (3) | Compodiff (Gu et al., 2023) | CLIP ViT-L | $\langle I_r, T_c, I_t \rangle$ | 55.0 | 72.6 | 13.4 | 15.8 | 36.0 | 48.6 |
| | CoVR (Ventura et al., 2024) | BLIP ViT-L | $\langle I_r, T_c, I_t \rangle$ | 68.2 | 78.9 | - | - | 27.7 | 44.6 |
| | CASE (Levy et al., 2024) | BLIP ViT-L | $\langle I_r, T_c, I_t \rangle$ | 65.8 | 78.5 | - | - | - | - |
| (4) | CIReVL (Karthik et al., 2023) | CLIP ViT-L | ✗ | 52.3 | 64.9 | 19.0 | 20.9 | 28.6 | 48.6 |

in LinCIR (Gu et al., 2024). Considering the strong performance and practical advantages (efficient training and inference), we believe RTD stands for a promising direction in the CIR domain.

## 4.4 ABLATION STUDIES

Table 6 presents the effectiveness of the proposed components: target-anchored text contrastive loss (TCL), refined batch sampling (RB), and refined concatenation scheme (RC). All evaluation results are on the validation splits. All model variants use ViT-L/14 and a projection module $\phi$ from Lin-CIR, making the results in row 1 indicative of the original performance of LinCIR. We first compare the impact of the text pairs fed into TCL loss. We compare our design choice $(T_{r+c}, T_t)$ (from the generated text triplets) with $(T_r, T_r)$, which is the sole option for constructing a pair given a single conventional caption $T_r$. The results demonstrate that, on average, using generated triplets (3rd row) is more effective than using original conventional text pairs (2nd row), particularly in the CIRR and CIRCO datasets. In addition, RB (4th row) and RC (6th row) significantly enhance the overall performance, demonstrating the effectiveness of these components. Finally, we measure the impact of using the frozen text encoder for target caption $T_t$, denoted as "Anchor" in the table. Significant performance degradation is observed when the learnable text encoder is used for extracting the embedding of the target caption $T_t$ (5th row) compared to the target-anchored case (4th row), supporting the importance of the anchoring design choice.

Table 6: **Ablation study.** Unlike in Tables 2 to 5, for ablation studies and analyses, validation splits of three CIR datasets are used for evaluation. We measure the impact of TCL loss (Eq. (1)), refined batch sampling (RB), and refined concatenation scheme (RC). All models are based on LinCIR ViT-L/14. The first row denotes the vanilla LinCIR without RTD.

| TCL | | RB | RC | CIRR | | CIRCO | | FashionIQ | | Avg |
|---|---|---|---|---|---|---|---|---|---|---|
| Text pair | Anchor | | | R@5 | R@10 | mAP@10 | mAP@25 | R@10 | R@50 | |
| - | - | ✗ | ✗ | 54.29 | 67.76 | 12.67 | 14.45 | 27.42 | 47.71 | 37.38 |
| $(T_r, T_r)$ | ✔ | ✗ | ✗ | 55.99 | 69.72 | 13.40 | 15.18 | 28.16 | 48.82 | 38.54 |
| $(T_{r+c}, T_t)$ | ✔ | ✗ | ✗ | **58.19** | **71.54** | 14.36 | 16.03 | 26.93 | 47.94 | 39.17 |
| $(T_{r+c}, T_t)$ | ✔ | ✔ | ✗ | **58.19** | 71.27 | 14.96 | 16.67 | 27.42 | 49.33 | 39.64 |
| $(T_{r+c}, T_t)$ | ✗ | ✔ | ✗ | 54.34 | 66.97 | 12.23 | 13.64 | 25.02 | 45.31 | 36.25 |
| $(T_{r+c}, T_t)$ | ✔ | ✔ | ✔ | 57.90 | 71.13 | **16.10** | **17.84** | **30.24** | **51.08** | **40.72** |

## 4.5 MORE ANALYSES ON RTD

Here, we show more analyses on RTD with the same setting to Table 6. Namely, we use the same evaluation dataset, ViT-L/14 CLIP backbone, and a projection module $\phi$ from LinCIR.

**[Impact of the text triplet generation strategies]** As explained in Section 3.1, we evaluate RTD using both 1) publicly available LLM-based text triplets (from IP2P, Compodiff, CoVR, and CASE) along with efficient in-context learning-based text triplets, and 2) LLM-free, rule-based triplets.

Table 7: **The effectiveness of different types of text triplets for RTD.** "Efficient in-context learning" denotes an efficient implementation using in-context learning with LLaMA3-8B, without a fine-tuning phase. Details and examples of each text triplet dataset are summarized in Table A.1 and Table A.2, respectively. Other details are the same as Table 6.

| Method | Text Triplet Datasets | Use LLM | CIRR | | CIRCO | | FashionIQ | | Avg |
|---|---|---|---|---|---|---|---|---|---|
| | | | R@5 | R@10 | mAP@10 | mAP@25 | R@10 | R@50 | |
| LinCIR | - | - | 54.29 | 67.76 | 12.67 | 14.45 | 27.42 | 47.71 | 37.38 |
| +RTD | Rule-based | ✗ | 56.71 | 70.34 | 15.01 | 16.98 | 30.37 | 51.94 | 40.23 **(+ 2.85)** |
| | IP2P (Brooks et al., 2023) | ✔ | 58.65 | 71.59 | 15.94 | 17.97 | 29.62 | 50.67 | 40.67 **(+ 3.29)** |
| | Compodiff (Gu et al., 2023) | ✔ | 57.90 | 71.13 | 16.10 | 17.84 | 30.24 | 51.08 | 40.72 **(+ 3.34)** |
| | Efficient in-context learning | ✔ | 59.27 | 71.78 | 15.81 | 17.45 | 29.69 | 51.44 | 40.91 **(+ 3.53)** |
| | CoVR (Ventura et al., 2024) | ✔ | 59.82 | 72.64 | 15.35 | 17.01 | 29.58 | 50.79 | 41.86 **(+ 4.48)** |
| | CASE (Levy et al., 2024) | ✔ | 56.28 | 69.29 | 11.13 | 12.66 | 26.63 | 47.78 | 37.69 **(+ 0.31)** |

Table 8: **Impact of the update scheme.** Two update schemes are compared: (1) using the original objective from baseline and (2) using RTD. For a fair comparison, in both schemes, $\phi$ is updated first and $\psi_T$ is updated top on the frozen modules. Other details are the same as Table 6.

| | CIRR | | CIRCO | | FashionIQ | | Avg |
|---|---|---|---|---|---|---|---|
| | R@5 | R@10 | mAP@10 | mAP@25 | R@10 | R@50 | |
| Baseline(Pic2Word) | 51.40 | 64.43 | 8.77 | **10.12** | 25.34 | 44.88 | 32.15 |
| +naïve tuning | 19.21 | 27.51 | 1.29 | 1.61 | 4.4 | 11.15 | 10.86 |
| + RTD | **56.64** | **69.77** | **8.83** | 9.81 | **27.59** | **48.90** | **36.92** |
| Baseline(LinCIR) | 54.29 | 67.76 | 12.67 | 14.45 | 27.42 | 47.71 | 36.86 |
| +naïve tuning | 52.67 | 66.78 | 11.40 | 12.99 | 26.34 | 45.92 | 35.52 |
| + RTD | **57.90** | **71.13** | **16.10** | **17.84** | **30.24** | **51.08** | **40.72** |

Table 7 shows that RTD consistently improves ZS-CIR performance across them (+3.29 for IP2P, +3.34 for Compodiff, +3.53 for in-context learning, +4.48 for CoVR, and +0.31 for CASE, and rule-based triplets achieve 2.85, respectively). We believe this result demonstrates the reproducibility and consistency of RTD, with the rule-based triplets performing comparably to LLM-generated ones, indicating that efficient rule-based triplets are sufficient to achieve strong ZS-CIR performance. The marginal improvement in CASE is largely due to the poor quality of text triplets resulting from its construction pipeline that prioritizes CIR triplet quality over text triplet quality, as shown in Table A.1. Further details can be found in Appendix A.2, and additional analyses, including data scales related to text triplets, are provided in Appendix B.4.

**[Impact of the update scheme for the text encoder]** To verify that our improvements cannot be achievable solely by tuning the text encoder backbone without considering the task discrepancy, we additionally measure the results of previous methods (Pic2Word and LinCIR) when naively updating text encoders. Namely, after training $\phi$ while keeping all other networks frozen as in previous methods, we additionally update the text encoder using the original loss function, while fixing other modules including $\phi$. We denote this update rule as "naïve tuning" in the Table 8. Unlike RTD, we observe that just naively updating the text encoder ("naïve tuning") significantly degrades the performance of the baseline. The results indicate that merely updating the text backbone is not beneficial for ZS-CIR; instead, mitigating task discrepancy through RTD is necessary.

## 5 CONCLUSION

Our research presents RTD, a novel post-processing approach that is easily integrable into existing projection-based ZS-CIR methods, aimed at enhancing text encoder capabilities. By leveraging easily obtainable text triplets, RTD addresses the challenges posed by task discrepancies in these ZS-CIR methods. Empirical evaluations demonstrate that RTD significantly boosts the performance of existing projection-based ZS-CIR methods across diverse datasets and model backbones, competing with or outperforming other state-of-the-art CIR methods beyond projection-based approaches with much greater efficiency, underscoring its effectiveness and versatility.

## 6 REPRODUCIBILITY STATEMENT

We provide all necessary details for reproduction in the manuscript, including implementation details, metrics, datasets, and baselines, as described in Section 4.1. Additionally, the training and evaluation dataset details are elaborated in Appendix A.1 and Appendix A.2. The anonymized code for reproducing our results is provided in the supplementary material.

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

## A  ADDITIONAL IMPLEMENTATION DETAILS

### A.1  CIR DATASETS

FashionIQ (Wu et al., 2021) is a dataset that contains fashion-related images from three main cate-gories: Shirts, Dresses, and Toptee. It has a total of 30,134 triplets, which were created from 77,684 images. As the ground truth labels are not publicly available, we utilize the results from the vali-dation set for our analysis and comparison. CIRR (Liu et al., 2021) encompasses a wider range of domains and contains images with more complex descriptions compared to FashionIQ. It contains 36,554 triplets extracted from 21,552 images, which are sourced from the well-known NLVR2 nat-ural language inference dataset (Suhr et al., 2018). As pointed out in previous works (Saito et al., 2023; Gu et al., 2024; Baldrati et al., 2023), CIRR and FashionIQ suffer from a significant number of false negatives, which can potentially lead to inaccurate retrieval evaluations (Baldrati et al., 2023; Saito et al., 2023). To address this issue, CIRCO (Baldrati et al., 2023), based on COCO images (Lin et al., 2014), is recently introduced by providing multiple positive images for each query. This approach enables a more reliable and robust mAP metric (Musgrave et al., 2020; Chun et al., 2022), which is essential for accurate evaluation of retrieval performance.

We additionally provide results on two more benchmark datasets, GeneCIS (Vaze et al., 2023) and COCO Object Composition introduced by (Saito et al., 2023), in Appendix B.1. GeneCIS (Vaze et al., 2023) is also constructed based on COCO images and the Visual Attributes in the Wild dataset (Pham et al., 2021). GeneCIS introduces four task variations: (1) focus on an attribute, (2) change an attribute (3) focus on an object and (4) change an object. These tasks explore different aspects of image retrieval and manipulation. For the COCO Object Composition task, we utilize 5000 images from the COCO validation dataset to evaluate object composition. Our objective is to retrieve an image that contains an object specified by a query image, along with scenes or objects described using text. The composed query is constructed by combining "a photo of [\$], $[obj_1]$, $[obj_2]$ ... and $[obj_n]$" where $[obj_i]$ are text descriptions.

### A.2  DETAILS OF TEXT TRIPLETS

Here, we describe the details of the LLM-based and rule-based text triplet generation process. As shown in Figure D.2 and D.3, which showcases examples of both LLM-based and rule-based triplets, both approaches produce natural and coherent text triplets. Note that none of the datasets used for generating text triplets overlap with the data used in the target CIR benchmarks, with the exception of the CASE dataset (Levy et al., 2024). The source of the CASE dataset is VQA2.0 (Goyal et al., 2017), which is constructed from the COCO dataset (Lin et al., 2014), potentially leading to overlap in cases involving COCO object composition (Saito et al., 2023).

**[Detailed explanation on LLM-based triplets]** As described in Section 3, besides Compodiff (Gu et al., 2023), we conduct experiments using various publicly available text triplets: IP2P (Brooks et al., 2023), COVR (Ventura et al., 2024), and CASE (Levy et al., 2024). Although the primary objective of these approaches is to generate CIR triplets $(I_r, T_c, I_t)$, they also produce text triplets. Below, we provide detailed descriptions of how text triplets are constructed in each approach (Note again that their final product is CIR triplets). There are two main ways to generate text triplets using LLMs: 1) generating both conditional text $T_c$ and target caption $T_t$ given reference caption $T_r$ using fine-tuned LLM for this task, such as IP2P, Compodiff; and 2) generating only conditional text $T_t$ given pairs $(T_r, T_t)$ from pre-existing captions by identifying with visually or text semantically similar such as CoVR (Ventura et al., 2024), and CASE (Levy et al., 2024). In addition to these existing datasets, we implement an efficient in-context learning-based generation process. Examples and summaries of each dataset can be found in Table A.1 and Table A.2.

**IP2P** employs GPT-3 for text triplets generation and fine-tunes it with a human-curated small set of 700 text triplets. Namely, given reference captions $T_r$ sampled from LAION-Aesthetics V2 6.5+ dataset (Schuhmann et al., 2022a), the corresponding conditional texts $T_c$ and corresponding target captions $T_t$ are manually written by humans. After fine-tuning on this small set of text triplets, the model generates 454k text triplets: reference captions $T_r$ from the LAION-Aesthetics V2 6.5+ dataset (Schuhmann et al., 2022a) are provided as input to the fine-tuned LLM, whose output predicts the corresponding conditioning text $T_c$ and the target caption $T_t$. Note that the LAION-Aesthetics dataset is not related to the original source datasets (FashionIQ, NLVR2, and MS-COCO) used

in existing CIR benchmarks (FashionIQ, CIRR, and CIRCO), ensuring no overlap with the CIR benchmarks.

**Compodiff** enhances the scalability of the IP2P text triplet generation process by modifying the choice of LLM and expanding the fine-tuning dataset. As described in [Gu et al. (2023), Section 4], the OPT-6.7B model is utilized and fine-tuned with LoRA on the above 454k text triplets of IP2P (Brooks et al., 2023). Then, similar to the IP2P approach, given reference captions from the LAION dataset (Schuhmann et al., 2022b), fine-tuned LLM generates the corresponding conditioning texts and target captions.

**COVR** starts by identifying similar caption pairs from the WebVid2M dataset (Bain et al., 2021), which contains 2.5 million video-caption pairs. These pairs serve as the reference captions ($T_r$) and target captions ($T_t$). Then, given these pairs ($T_r, T_t$), LLM generates conditional captions that describe the differences between the paired captions. The LLaMA-7B model (Touvron et al., 2023) is utilized for this purpose and is fine-tuned on an expanded version of the above 700 manually annotated triplets used in IP2P (adding 15 annotations for more diverse cases).

**CASE** uses VQA2.0 dataset (Goyal et al., 2017), which consists of (image, question, answer) triplets. Given $(I, Q, A)$ triplets, complementary triplets $(I_c, Q, A_c)$ are manually selected based on visually similar image $I_c$ with three rules: 1) the premise assumed in question $Q$ holds for both $I$ and $I_c$, 2) $Q$ is logical for $I_c$, and 3) the answer $A_c$ for $I_c$ differs from $A$. Then, conditional text $T_c$ is generated by GPT-3, describing differences between image pairs $(I, I_c)$ without fine-tuning, leveraging in-context learning with a few examples. Since the VQA2.0 dataset is derived from the COCO dataset, COCO captions that match VQA2.0 images are used to form text triplets.

As seen in Table A.1, compared to other approaches, the quality of the relationships between $T_r$, $T_c$, and $T_t$ is not always satisfactory, which results in minimal performance gain as shown in Table 7. Namely, unlike other CIR datasets that first create high-quality text triplets before generating CIR triplets, CASE generates the conditioning text $T_c$ using the reference image $I_r$ and target image $I_t$. The provided reference text $T_r$ and target text $T_t$ are taken directly from the captions of reference image $I_r$ and target image $I_t$ in the VQA2.0 dataset. Therefore, due to the poor descriptiveness of these captions and their lack of consideration for the conditioning text $T_c$, while $T_c$ can effectively explain the visual differences between $I_r$ and $I_t$, it often fails to capture the differences between $T_r$ and $T_t$ adequately.

**Efficient in-context learning** refers to our efficient implementation which uses a recent and powerful LLM, LLaMA3-8B (Dubey et al., 2024). This approach performs in-context learning using reference captions $T_r$ from the CC3M dataset (Sharma et al., 2018), guided by a custom-designed prompt with a few examples of textual modifications (*e.g.*, replace, change, remove, ...). Specifically, given a reference caption $T_r$, the prompt instructs the model to generate a target caption $T_t$, which is a complete sentence that slightly differs from the corresponding reference caption. Then, the prompt guides the model to generate conditioning text that explains the differences between $T_r$ and $T_t$, based on the above pre-defined textual modifications. Compared to Compodiff, which takes 3.8 hours to generate 1 million text triplets, this version requires only 1.5 hours. In Table 7, we verify that this more efficient version achieves competitive performance compared to the other fine-tuned LLM approaches.

Table A.1: **Examples of text triplet datasets.**

| Text triplets | Reference text $T_r$ | Conditioning text $T_c$ | Target text $T_t$ |
|---|---|---|---|
| Rule-based | "another wall at my home" | "bedroom is added in place of wall" | "another bedroom at my home" |
| IP2P (Brooks et al., 2023) | "watercolor of your pet!" | "make it a huge grizzly bear" | "watercolor of a huge grizzly bear!" |
| Compodiff (Gu et al., 2023) | "Chinese landscape watercolor painting" | "make the landscape a cityscape" | "chinese cityscape watercolor painting" |
| Efficient in-context learning | "young business woman on a bench" | "add a laptop" | "young business woman on a bench with a laptop" |
| CoVR (Ventura et al., 2024) | "Two little boys are running" | "Have them dance" | "Two little boys are dancing" |
| CASE (Levy et al., 2024) | "A scone with an orange slice on a plate" | "This food is not acidic" | "a close up of a muffin on a plate on a table" |

**[Detailed explanation on rule-based triplets]** To construct rule-based triplets, we mainly follow the process described in [Gu et al. (2023), Section 4.1]. Firstly, given reference captions, important keywords like nouns are extracted with a part-of-speech (POS) tagger via the Spacy library. Then, the keyword is filtered by frequency filtering with hard thresholding to focus only on frequently occurring keywords. Specifically, we only use keywords that appear more than 100. After applying keyword frequency filtering, the remaining keyword list is used to create caption triplets $(T_r, T_c, T_t)$. To generate text triplets, a keyword from the given $T_r$ is selected, and alternative keywords are

Table A.2: **Summaries of text triplet dataset.**

| Dataset | Use LLM | Model | Fine-tuning strategy | # of text triplets |
|---|---|---|---|---|
| Rule-based | ✗ | ✗ | ✗ | 1.3M |
| IP2P (Brooks et al., 2023) | ✔ | GPT-3 | Fine-tuned on 700 human-written text triplets | 450K |
| Compodiff (Gu et al., 2023) | ✔ | OPT-6.7B | Fine-tuned on 450k IP2P text triplets | 2.5M |
| Efficient in-context learning | ✔ | LLaMA3-8B | In-context learning | 1M |
| CoVR (Ventura et al., 2024) | ✔ | LLaMA-7B | Fine-tuned on 700 human-written text triplets | 700K |
| CASE (Levy et al., 2024) | ✔ | GPT-3 | In-context learning | 350K |

extracted based on text similarity scores ranging from 0.5 to 0.7, using the SBERT all-MiniLM-L6-v2 (Reimers & Gurevych, 2019). The target caption $T_t$ is then constructed by substituting the original keyword with a similar alternative. The conditioning text $T_c$ is generated using randomly selected pre-defined templates, as detailed in Table D.1. Here, most of the templates are similar to that of Compodiff (Gu et al., 2023). We use captions from the CC3M dataset (Sharma et al., 2018) as reference captions $T_r$. Note that CC3M is not related to the existing CIR benchmarks, which again ensures no overlap with the CIR benchmarks.

Since the quality of the generated triplets with the above procedure may not be optimal, we employ an additional filtering process. Compodiff (Gu et al., 2023) employs an additional filtering process that uses cosine similarities between generated images and texts, calculated by CLIP encoders. However, as we do not have images for captions, we filter the inappropriate texts using only textual information inspired by LinCIR (Gu et al., 2024). Namely, we calculate the similarity between the CLIP text embedding of $T_t$ and the CLIP text embedding of "a photo of [\$]" where [\$] is obtained by $T_t$ projected by $\phi$ from LinCIR (ViT-L/14). Following LinCIR noise (Unif$(0, 1) \times \mathcal{N}(0, 1)$) is injected before passing through $\phi$. After calculating the above similarity, texts whose similarities are less than the threshold (0.75) are removed. The same process is also applied to the reference caption $T_r$ and the intersection of filtering processes for $T_t$ and $T_r$ is used for the final dataset whose size becomes 1.3M. As described in Appendix B.5, we verify that this filtering process is effective. However, this does not imply that the effectiveness of rule-based text triplets is solely dependent on the use of a projection module in the filtering process; even without filtering, the enhancement from RTD remains significant.

## B    ADDITIONAL EXPERIMENTAL RESULTS

### B.1    RESULTS ON GENECIS (VAZE ET AL., 2023) AND COCO OBJECT COMPOSITION

We observe that incorporating our approach with ZS-CIR methods leads to marginal but consistent performance improvements on GeneCIS as shown in Table B.1. The relatively smaller performance difference compared to other datasets can be attributed to the discrepancy between the format of the conditioning text of GeneCIS and the ZS-CIR methods training methodology. Namely, GeneCIS only uses the fixed four text instructions "change attribute", "focus attribute", "change object" and "focus object", which is different from the usual text instruction we expected (*e.g.*, "change the dog to a cat").

In the experiment on COCO object composition, we observe a significant performance improvement, similar to the results obtained on other datasets in Table B.2. This finding reaffirms that our approach, when combined with ZS-CIR methods, consistently achieves strong performance, demonstrating its generalizability.

### B.2    RESULTS ON LARGER BACKBONE (VIT-G/14)

We further evaluate the performance of RTD using the significantly larger backbone (OpenCLIP ViT-G/14 (Ilharco et al., 2021)). As described in Section 4.4, we use the projection module $\phi$ from LinCIR (Gu et al., 2024). Since the pre-trained projection module $\phi$ for LinCIR (Gu et al., 2024) (ViT-G/14) is not publicly available, we reproduce it and integrate RTD with it. We emphasize that similar to our previous results, RTD again achieves remarkable gains across all datasets. Here, we set the learning rate as $10^{-6}$.

Table B.1: GeneCIS results

|  |  | Average | | |
|  |  | R@1 | R@2 | R@3 |
| --- | --- | --- | --- | --- |
| ViT-B | Pic2Word | 11.13 | 21.08 | 31.05 |
|  | +RTD | 12.03 (+**0.90**) | 21.61 (+**0.53**) | 31.09 (+**0.04**) |
|  | SEARLE | 12.19 | 22.56 | 32.03 |
|  | +ours | 12.82 (+**0.63**) | 22.97 (+**0.41**) | 32.44 (+**0.41**) |
|  | LinCIR | 12.23 | 21.29 | 30.80 |
|  | +ours | 12.83 (+**0.60**) | 22.83 (+**1.54**) | 32.22 (+**1.42**) |
| ViT-L | Pic2Word | 11.18 | 21.45 | 30.55 |
|  | +ours | 11.92 (+**0.74**) | 22.32 (+**0.87**) | 31.33 (+**0.78**) |
|  | SEARLE | 12.30 | 22.08 | 31.29 |
|  | +ours | 12.40 (+**0.10**) | 22.82 (+**0.74**) | 32.37 (+**1.08**) |
|  | LinCIR | 12.45 | 22.66 | 32.06 |
|  | +ours | 13.18 (+**0.73**) | 23.12 (+**0.46**) | 32.77 (+**0.71**) |

Table B.2: COCO object composition results

|  |  | COCO | | |
|  |  | R@1 | R@5 | R@10 |
| --- | --- | --- | --- | --- |
| ViT-B | Pic2Word | 6.88 | 13.6 | 17.52 |
|  | +ours | 7.62 (+**0.74**) | 20.23 (+**6.63**) | 28.69 (+**11.17**) |
|  | SEARLE | 9.52 | 21.45 | 29.38 |
|  | +ours | 11.01 (+**1.49**) | 24.34 (+**2.89**) | 32.84 (+**3.46**) |
|  | LinCIR | 7.15 | 18.38 | 27.3 |
|  | +ours | 9.59 (+**2.44**) | 21.66 (+**3.28**) | 30.66 (+**3.36**) |
| ViT-L | Pic2Word | 10.26 | 23.67 | 32.53 |
|  | +ours | 10.26 (+**0.00**) | 24.66 (+**0.99**) | 33.56 (+**1.03**) |
|  | SEARLE | 12.07 | 26.13 | 35.17 |
|  | +ours | 14.38 (+**2.31**) | 29.74 (+**3.61**) | 38.09 (+**2.92**) |
|  | LinCIR | 11.37 | 24.53 | 33.85 |
|  | +ours | 14.6 (+**3.23**) | 29.84 (+**5.31**) | 38.87 (+**5.02**) |

Table B.3: FashionIQ results on larger OpenCLIP ViT-G/14 backbone (Ilharco et al., 2021).

| Method | Shirt | | Dress | | Toptee | | Average | |
|  | R@10 | R@50 | R@10 | R@50 | R@10 | R@50 | R@10 | R@50 |
| --- | --- | --- | --- | --- | --- | --- | --- | --- |
| LinCIR (reported in Gu et al. (2024)) | 46.76 | 65.11 | 38.08 | 60.88 | 50.48 | 71.09 | 45.11 | 65.69 |
| LinCIR (reproduced) | 46.61 | 64.72 | 38.18 | 60.54 | 49.26 | 70.83 | 44.68 | 65.36 |
| +RTD | 47.20 (+**0.59**) | 66.24 (+**1.52**) | 39.86 (+**1.68**) | 63.01 (+**2.47**) | 51.56 (+**2.30**) | 72.51 (+**1.68**) | 46.21 (+**1.54**) | 67.26 (+**1.90**) |

Table B.4: CIRR and CIRCO results on larger OpenCLIP ViT-G/14 backbone (Ilharco et al., 2021).

| ViT-G | CIRR | | | CIRCO | | | |
|  | R@1 | R@5 | R@10 | mAP@5 | mAP@10 | mAP@25 | mAP@50 |
| --- | --- | --- | --- | --- | --- | --- | --- |
| LinCIR (reported in Gu et al. (2024)) | 35.25 | 64.72 | 76.05 | 19.81 | 21.01 | 23.03 | 24.18 |
| LinCIR (reproduced) | 34.94 | 64.51 | 76.12 | 20.63 | 21.93 | 24.12 | 25.20 |
| +RTD | 36.31 (+**1.37**) | 67.47 (+**2.96**) | 78.31 (+**2.19**) | 21.08 (+**0.45**) | 22.29 (+**0.36**) | 24.46 (+**0.34**) | 25.44 (+**0.24**) |

## B.3 MORE EFFICIENT VARIANTS

Table B.5 presents the results of the more efficient implementations of our approach in terms of the number of updated parameters. Specifically, instead of updating the entire set of parameters of the text encoder, we explore updating only a few layers of the network when applying RTD, Our findings indicate that updating only the fully connected layers (denoted as "Whole FCs") nearly matches the performance of the full model while using less than half the number of learnable parameters (40.72 vs. 40.53 average score). Additionally, we verify that updating only three fully connected layers, whose parameter size matches the projection module $\phi$ and constitutes 11.5% of the full model, is also sufficiently effective. We test various three-layer updating strategies: "First 3 FCs": the first three layers (closest to the input), "middle 3 FCs": the middle three layers, "Last 3 FCs": the last three layers, and "Interleave 3 FCs": an interleaved selection of three layers (first, middle, and last

Table B.5: **More efficient variants.** "Learnable params (%)" denotes the percentage of learnable parameters relative to the entire set of parameters in the text encoder.

| | | CIRR | | CIRCO | | FashionIQ | | Avg |
|---|---|---|---|---|---|---|---|---|
| Training variants | Learnable params (%) | R@5 | R@10 | mAP@10 | mAP@25 | R@10 | R@50 | |
| Baseline(LinCIR) | 0% | 54.29 | 67.76 | 12.67 | 14.45 | 27.42 | 47.71 | 37.38 |
| +RTD (Full model) | 100% | 57.90 | 71.13 | 16.10 | 17.84 | 30.24 | 51.08 | 40.72 |
| +RTD (Whole FCs) | 45.8% | 57.76 | 71.35 | 15.03 | 16.90 | 30.31 | 51.81 | 40.53 |
| +RTD (Front 3 FCs) | 11.5% | 55.65 | 69.83 | 13.95 | 15.81 | 28.69 | 49.92 | 38.98 |
| +RTD (Middle 3 FCs) | 11.5% | 56.69 | 70.03 | 14.66 | 16.58 | 28.55 | 49.84 | 39.39 |
| +RTD (Last 3 FCs) | 11.5% | 56.84 | 69.74 | 14.81 | 16.70 | 29.16 | 50.43 | 39.61 |
| +RTD (Interleave 3 FCs) | 11.5% | 57.21 | 70.65 | 15.20 | 17.13 | 28.91 | 50.17 | 39.88 |

layers). Among these, we verify that the "Interleave 3 FCs" shows the best result, maintaining competitive performance with the full model (40.72 vs. 39.88 average score). We believe these findings suggest a promising direction for enhancing the training efficiency of our approach by selectively updating only specific layers of the text encoder.

## B.4 EFFECTIVENESS OF RTD ACROSS DATASET SCALES

We conduct experiments with various scales of training text triplets. For the small-scale text triplets, we sub-sampled text triplets from LLM-based text triplets. Thus, the last row in the Table B.6 denotes the original result (LLM-based RTD result). We also measure the effectiveness of RTD using large-scale text triplets (up to 5M) by combining publicly available text triplets (IP2P, CoVR) with ours (LLM-based, rule-based). Here, validation splits of all three benchmark datasets are utilized and full results will be included in the final version.

As shown in Table B.6 and B.7, we confirm that small-scale text triplets are sufficient to achieve the effectiveness of RTD. We believe the main reason for this is that, to reduce task discrepancy, only the relationship between the concatenated caption (reference caption + conditioning caption, $T_{r+c}$) and the target caption $T_t$ needs to be learned. We believe this learning task requires much less data compared to learning representations from scratch. Moreover, since the text encoder is already pre-trained, the model does not need significant changes to learn this simple but crucial learning task for CIR.

Table B.6: **Results across different scales of LLM-based text triplets**. In each row, text triplets are sub-sampled from 2.5M original LLM-based text triplets provided by Compodiff (Gu et al., 2023)

| # of triplets | CIRR R@5 | CIRCO mAP@10 | FashionIQ R@10 | Avg |
|---|---|---|---|---|
| 1K | 56.64 | 15.66 | 29.89 | 34.06 |
| 50K | 57.40 | 15.95 | 30.77 | 34.71 |
| 100K | 57.16 | 16.03 | 30.57 | 34.59 |
| 2.5M | 57.90 | 16.10 | 30.24 | 34.75 |

Table B.7: Results of larger-sized text triplets

| IP2P | CoVR | Compodiff | Template-based | CIRR R@5 | CIRCO mAP@10 | FashionIQ R@10 | Avg | # of triplets |
|---|---|---|---|---|---|---|---|---|
| ✔ | | | | 58.65 | 15.94 | 29.62 | 34.74 | 450k |
| | ✔ | | | 59.82 | 15.35 | 29.58 | 34.92 | 700k |
| | | ✔ | | 57.90 | 16.10 | 30.24 | 34.75 | 2.5M |
| | | | ✔ | 56.71 | 15.01 | 30.37 | 34.03 | 1.3M |
| ✔ | ✔ | | | 59.32 | 16.10 | 30.81 | 35.41 | 1.25M |
| ✔ | ✔ | ✔ | | 59.08 | 16.15 | 30.97 | 35.40 | 3.75M |
| ✔ | ✔ | ✔ | ✔ | 58.65 | 16.54 | 31.22 | 35.47 | 5.05M |

## B.5 ABLATIONS ON FILTERING PROCESS

In rule-based text triplet generation, we highlight that the filtering process using the projection module from LinCIR is *marginally effective*. As demonstrated in the Table B.8, even without the filtering

procedure, the enhancement of RTD from LinCIR remains considerable. This result demonstrates that the effectiveness of our rule-based text triplets is not solely dependent on the use of the projection module from LinCIR in the filtering process.

Table B.8: Ablations on filtering process

| Type | LinCIR-based filtering | CIRR R@5 | CIRCO mAP@10 | FashionIQ R@10 | Avg |
|---|---|---|---|---|---|
| LinCIR | - | 54.29 | 12.67 | 27.42 | 31.46 |
| +RTD (rule-based) | ✗ | 55.49 | 14.75 | 30.24 | 33.49 |
| +RTD (rule-based) | ✔ | 56.71 | 15.01 | 30.37 | 34.03 |

## B.6 ABLATIONS ON NOISE INJECTION

We conduct an ablation study of the different noise types employed for the "refined concatenation scheme" shown in Figure 2. We compare three different noise types, uniform distribution, Gaussian distribution, and LinCIR-ish noise ($\text{Unif}(0,1) \times \mathcal{N}(0,1)$). We also examine the scale of LinCIR-ish noise from 0.1, 0.5, and 1. We report the test scores for CIRR and CIRCO, as well as the FashionIQ validation scores for Pic2Word, SEARLE, and LinCIR in Table B.9 and Table B.10. In the tables, we observe that all noise distributions show decent performance and LinCIR-like noises show slightly better performances than uniform distribution and normal distribution. We also observe that the different scale choice for the LinCIR-like noise somewhat affects the overall performances. In the main experiments, we choose 0.5 for the noise scale, following the observed performance improvements.

Table B.9: Noise type variation on CIRR/CIRCO dataset

| | | Noise type | Scale | CIRR R@1 | R@5 | R@10 | CIRCO mAP@5 | mAP@10 | mAP@25 | mAP@50 |
|---|---|---|---|---|---|---|---|---|---|---|
| | Pic2Word | - | - | 13.64 | 37.45 | 52.22 | 2.85 | 3.24 | 3.89 | 4.31 |
| | | Unif(-1,1) | 1 | 23.23 | 50.55 | 64.28 | 4.29 | 4.57 | 5.19 | 5.57 |
| | | $\mathcal{N}(0,1)$ | 1 | 21.18 | 47.78 | 61.47 | 4.09 | 4.26 | 4.83 | 5.17 |
| | +ours | $\mathcal{N}(0,1) \times \text{Unif}(0,1)$ | 0.1 | 23.52 | 51.13 | 64.53 | 5.13 | 5.46 | 6.17 | 6.62 |
| | | $\mathcal{N}(0,1) \times \text{Unif}(0,1)$ | 0.5 | 23.01 | 51.18 | 64.84 | 4.29 | 4.57 | 5.19 | 5.57 |
| | | $\mathcal{N}(0,1) \times \text{Unif}(0,1)$ | 1 | 23.59 | 51.76 | 65.16 | 6.39 | 6.66 | 7.64 | 8.16 |
| | SEARLE | - | - | 23.71 | 53.3 | 66.84 | 8.9 | 9.42 | 10.64 | 11.34 |
| | | Unif(-1,1) | 1 | 26.07 | 55.98 | 69.18 | 10.87 | 11.55 | 12.97 | 13.65 |
| | | $\mathcal{N}(0,1)$ | 1 | 26.41 | 56.68 | 69.47 | 10.91 | 11.53 | 12.88 | 13.6 |
| ViT-B/32 | +ours | $\mathcal{N}(0,1) \times \text{Unif}(0,1)$ | 0.1 | 26.02 | 55.47 | 68.15 | 10.43 | 11.07 | 12.37 | 13.07 |
| | | $\mathcal{N}(0,1) \times \text{Unif}(0,1)$ | 0.5 | 26.29 | 56.41 | 69.74 | 11.26 | 12.11 | 13.63 | 14.37 |
| | | $\mathcal{N}(0,1) \times \text{Unif}(0,1)$ | 1 | 26.43 | 56.58 | 69.76 | 11.42 | 12.04 | 13.38 | 14.1 |
| | LinCIR | - | - | 18.87 | 45.66 | 58.43 | 6.25 | 6.74 | 7.62 | 8.1 |
| | | Unif(-1,1) | 1 | 24.39 | 52.77 | 66.39 | 6.81 | 7.27 | 8.28 | 8.84 |
| | | $\mathcal{N}(0,1)$ | 1 | 24.63 | 53.52 | 66.63 | 7.6 | 7.97 | 8.92 | 9.49 |
| | +ours | $\mathcal{N}(0,1) \times \text{Unif}(0,1)$ | 0.1 | 24.58 | 53.3 | 66.65 | 9.6 | 10.11 | 11.47 | 12.15 |
| | | $\mathcal{N}(0,1) \times \text{Unif}(0,1)$ | 0.5 | 24.82 | 53.47 | 66.87 | 8.94 | 9.35 | 10.57 | 11.21 |
| | | $\mathcal{N}(0,1) \times \text{Unif}(0,1)$ | 1 | 25.4 | 54.58 | 67.69 | 8.17 | 8.53 | 9.72 | 10.35 |
| | Pic2Word | - | - | 24.22 | 51.49 | 64.05 | 8.27 | 9.1 | 10.09 | 10.75 |
| | | Unif(-1,1) | 1 | 28.24 | 55.95 | 68.77 | 8.14 | 8.81 | 9.83 | 10.37 |
| | | $\mathcal{N}(0,1)$ | 1 | 27.06 | 53.95 | 66.43 | 7.08 | 7.66 | 8.57 | 9.07 |
| | +ours | $\mathcal{N}(0,1) \times \text{Unif}(0,1)$ | 0.1 | 28.24 | 57.35 | 68.65 | 10.04 | 10.63 | 11.71 | 12.31 |
| | | $\mathcal{N}(0,1) \times \text{Unif}(0,1)$ | 0.5 | 27.86 | 56.24 | 68.48 | 9.13 | 9.63 | 10.68 | 11.27 |
| | | $\mathcal{N}(0,1) \times \text{Unif}(0,1)$ | 1 | 27.71 | 55.68 | 68.02 | 8.14 | 8.78 | 9.84 | 10.35 |
| | SEARLE | - | - | 24.89 | 52.31 | 65.69 | 11.62 | 12.72 | 14.33 | 15.13 |
| | | Unif(-1,1) | 1 | 26.96 | 56.99 | 69.52 | 15.82 | 16.78 | 18.54 | 19.39 |
| | | $\mathcal{N}(0,1)$ | 1 | 27.66 | 57.54 | 69.57 | 15.24 | 15.93 | 17.65 | 18.44 |
| ViT-L/14 | +ours | $\mathcal{N}(0,1) \times \text{Unif}(0,1)$ | 0.1 | 26.31 | 55.88 | 69.4 | 16.05 | 17.26 | 19.12 | 20.01 |
| | | $\mathcal{N}(0,1) \times \text{Unif}(0,1)$ | 0.5 | 27.04 | 56.82 | 69.95 | 16.53 | 17.89 | 19.77 | 20.68 |
| | | $\mathcal{N}(0,1) \times \text{Unif}(0,1)$ | 1 | 27.93 | 57.76 | 70.19 | 17.35 | 18.66 | 20.52 | 23.44 |
| | LinCIR | - | - | 23.76 | 52.89 | 66.46 | 13 | 14.11 | 15.81 | 16.68 |
| | | Unif(-1,1) | 1 | 26.58 | 56.31 | 68.94 | 17.23 | 18.2 | 20.11 | 21.03 |
| | | $\mathcal{N}(0,1)$ | 1 | 26.75 | 55.64 | 68.48 | 16.45 | 17.57 | 19.37 | 20.3 |
| | +ours | $\mathcal{N}(0,1) \times \text{Unif}(0,1)$ | 0.1 | 26.7 | 56.22 | 69.08 | 17.24 | 18.27 | 20.24 | 21.19 |
| | | $\mathcal{N}(0,1) \times \text{Unif}(0,1)$ | 0.5 | 26.63 | 56.17 | 68.96 | 17.11 | 18.11 | 20.06 | 21.01 |
| | | $\mathcal{N}(0,1) \times \text{Unif}(0,1)$ | 1 | 26.99 | 56.1 | 69.01 | 17.33 | 18.3 | 20.21 | 21.13 |

Table B.10: Noise type variation on FashionIQ dataset

| | | Noise type | Scale | Shirt R@10 | Shirt R@50 | Dress R@10 | Dress R@50 | Toptee R@10 | Toptee R@50 | Average R@10 | Average R@50 |
|---|---|---|---|---|---|---|---|---|---|---|---|
| ViT-B/32 | Pic2Word | - | - | 13.4 | 28.46 | 8.48 | 20.77 | 13.31 | 29.68 | 11.73 | 26.3 |
| | +ours | Unif(-1,1) | 1 | 21.84 | 37.63 | 18.49 | 39.61 | 23.0 | 43.91 | 21.11 | 40.38 |
| | | $\mathcal{N}(0,1)$ | 1 | 20.36 | 37.54 | 16.16 | 38.18 | 21.67 | 42.48 | 19.4 | 39.4 |
| | | $\mathcal{N}(0,1) \times$ Unif(0,1) | 0.1 | 22.23 | 39.35 | 19.98 | 41.7 | 23.81 | 45.23 | 22.01 | 42.09 |
| | | $\mathcal{N}(0,1) \times$ Unif(0,1) | 0.5 | 24.53 | 43.82 | 20.33 | 41.55 | 26.01 | 48.75 | 23.62 | 44.7 |
| | | $\mathcal{N}(0,1) \times$ Unif(0,1) | 1 | 23.06 | 40.48 | 20.33 | 41.75 | 24.12 | 46.35 | 22.5 | 42.86 |
| | SEARLE | - | - | 24.78 | 41.85 | 17.90 | 36.99 | 25.24 | 46.71 | 22.64 | 41.85 |
| | +ours | Unif(-1,1) | 1 | 23.75 | 42.25 | 20.18 | 40.36 | 25.14 | 46.46 | 23.02 | 43.02 |
| | | $\mathcal{N}(0,1)$ | 1 | 24.14 | 42.25 | 20.23 | 40.16 | 24.17 | 46.35 | 22.85 | 42.92 |
| | | $\mathcal{N}(0,1) \times$ Unif(0,1) | 0.1 | 25.12 | 44.85 | 20.92 | 41.40 | 26.57 | 47.63 | 24.20 | 44.62 |
| | | $\mathcal{N}(0,1) \times$ Unif(0,1) | 0.5 | 26.69 | 44.31 | 20.72 | 43.13 | 26.67 | 48.75 | 24.70 | 45.40 |
| | | $\mathcal{N}(0,1) \times$ Unif(0,1) | 1 | 25.07 | 44.01 | 20.43 | 41.00 | 26.11 | 47.12 | 23.87 | 44.04 |
| | LinCIR | - | - | 18.55 | 34.64 | 15.67 | 33.86 | 20.19 | 40.08 | 18.14 | 36.20 |
| | +ours | Unif(-1,1) | 1 | 21.79 | 39.35 | 18.89 | 40.21 | 23.66 | 45.33 | 21.45 | 41.63 |
| | | $\mathcal{N}(0,1)$ | 1 | 22.37 | 38.67 | 19.53 | 40.11 | 23.71 | 44.37 | 21.87 | 41.05 |
| | | $\mathcal{N}(0,1) \times$ Unif(0,1) | 0.1 | 23.95 | 44.11 | 19.83 | 41.99 | 26.62 | 47.58 | 23.47 | 44.56 |
| | | $\mathcal{N}(0,1) \times$ Unif(0,1) | 0.5 | 23.65 | 42.74 | 19.98 | 41.75 | 24.73 | 46.56 | 22.79 | 43.68 |
| | | $\mathcal{N}(0,1) \times$ Unif(0,1) | 1 | 22.82 | 41.12 | 19.78 | 41.70 | 25.09 | 47.07 | 22.56 | 43.29 |
| ViT-L/14 | Pic2Word | - | - | 26.59 | 42.93 | 21.32 | 43.53 | 28.10 | 48.19 | 25.34 | 44.88 |
| | +ours | Unif(-1,1) | 1 | 27.87 | 45.93 | 23.90 | 46.80 | 31.21 | 52.22 | 27.66 | 48.32 |
| | | $\mathcal{N}(0,1)$ | 1 | 26.94 | 44.95 | 23.45 | 45.56 | 30.34 | 51.45 | 26.91 | 47.32 |
| | | $\mathcal{N}(0,1) \times$ Unif(0,1) | 0.1 | 28.26 | 47.64 | 24.05 | 47.20 | 31.21 | 53.70 | 27.84 | 49.51 |
| | | $\mathcal{N}(0,1) \times$ Unif(0,1) | 0.5 | 27.97 | 46.96 | 23.50 | 46.65 | 31.31 | 53.09 | 27.59 | 48.90 |
| | | $\mathcal{N}(0,1) \times$ Unif(0,1) | 1 | 28.41 | 46.91 | 24.10 | 46.21 | 31.11 | 52.27 | 27.87 | 48.46 |
| | SEARLE | - | - | 26.94 | 45.34 | 19.58 | 40.80 | 28.45 | 49.77 | 24.99 | 45.30 |
| | +ours | Unif(-1,1) | 1 | 30.13 | 46.57 | 22.16 | 46.90 | 28.76 | 50.74 | 27.02 | 48.07 |
| | | $\mathcal{N}(0,1)$ | 1 | 26.99 | 43.23 | 21.17 | 44.82 | 27.54 | 49.06 | 25.23 | 45.70 |
| | | $\mathcal{N}(0,1) \times$ Unif(0,1) | 0.1 | 32.63 | 50.39 | 23.20 | 47.25 | 32.18 | 54.56 | 29.34 | 50.73 |
| | | $\mathcal{N}(0,1) \times$ Unif(0,1) | 0.5 | 31.80 | 49.31 | 23.20 | 47.30 | 31.41 | 54.00 | 28.80 | 50.20 |
| | | $\mathcal{N}(0,1) \times$ Unif(0,1) | 1 | 30.03 | 47.06 | 22.41 | 47.05 | 30.39 | 52.42 | 27.61 | 48.84 |
| | LinCIR | - | - | 30.42 | 47.99 | 21.86 | 44.77 | 29.98 | 50.38 | 27.42 | 47.71 |
| | +ours | Unif(-1,1) | 1 | 31.94 | 50.10 | 24.44 | 48.19 | 33.04 | 54.26 | 29.81 | 50.85 |
| | | $\mathcal{N}(0,1)$ | 1 | 31.70 | 49.41 | 23.90 | 48.19 | 33.23 | 53.54 | 29.27 | 50.38 |
| | | $\mathcal{N}(0,1) \times$ Unif(0,1) | 0.1 | 32.92 | 50.64 | 24.49 | 48.74 | 33.50 | 55.02 | 30.31 | 51.47 |
| | | $\mathcal{N}(0,1) \times$ Unif(0,1) | 0.5 | 32.83 | 50.44 | 24.49 | 48.24 | 33.40 | 54.56 | 30.24 | 51.08 |
| | | $\mathcal{N}(0,1) \times$ Unif(0,1) | 1 | 32.43 | 50.54 | 24.64 | 48.79 | 33.25 | 54.77 | 30.11 | 51.36 |

# C   TRAINING EFFICIENCY ANALYSIS

**[Generating text triplets cost]** Although generating text triplets is not our main contribution, for comprehensive understanding, we compare the generation time of LLM-based and rule-based approaches. Even when using LLMs, constructing text triplets is significantly more cost-effective than CIR triplets. Specifically, CIR triplets involve: 1) a subsequent, computationally intensive text-to-image generation phase (Brooks et al., 2023; Gu et al., 2023), or 2) the availability of image or video datasets along with an additional collection phase for semantically similar images or videos (Ventura et al., 2024; Levy et al., 2024). In contrast, generating text triplets bypasses these resource-heavy steps. For example, using 8 A100 GPUs, generating 1M text triplets takes 0.1 hours with the rule-based approach and 3.8 hours with the LLM-based approach from Compodiff (Gu et al., 2023) (OPT-6.7B). As described in Appendix A.2, a more efficient text triplet generation method using in-context learning with LLaMA3-8B reduces the generation time to 1.5 hours without the need for fine-tuning.

Therefore, while generating text triplets with LLMs incurs a higher cost compared to rule-based methods, it is still significantly faster (15 times) than generating CIR triplets (as used in CompoDiff), which utilize the text generation step as a preliminary phase for subsequent text-to-image generation. Thus, we believe LLM-based generation remains viable, but the rule-based approach is more efficient.

**[Additional training cost of RTD]** We believe the additional training cost of RTD is reasonably small: 0.5 hours using 8 A100 for CLIP ViT-L/14. This is reasonable compared to the original training times of projection-based ZS-CIR methods: LinCIR (0.5 hours), SEARLE (4.3 hours), and Pic2Word (3 hours). While our ablation studies are mainly conducted with LinCIR, RTD can be

integrated with other projection-based ZS-CIR methods, as shown in Table 2 and 3, at a similar additional cost. The small cost is largely due to the efficiency of updating the text encoder, which is significantly faster than updating the image encoder, resulting in 3.5 times faster inference time. Moreover, RTD can achieve strong performance with relatively few iterations (approximately 2000 iterations), as the text encoders are already pre-trained and only require minor adjustments to learn the relationships between text triplets.

## D  QUALITATIVE EXAMPLE ON CIRCO

We qualitatively illustrate the results of incorporating RTD into LinCIR on the CIRCO dataset in Figure D.1. The visual examples provide an intuitive demonstration of how the integration of RTD enhances the performance of LinCIR, effectively capturing the semantic meaning of the modification descriptions while preserving the relevant visual information from the reference image.

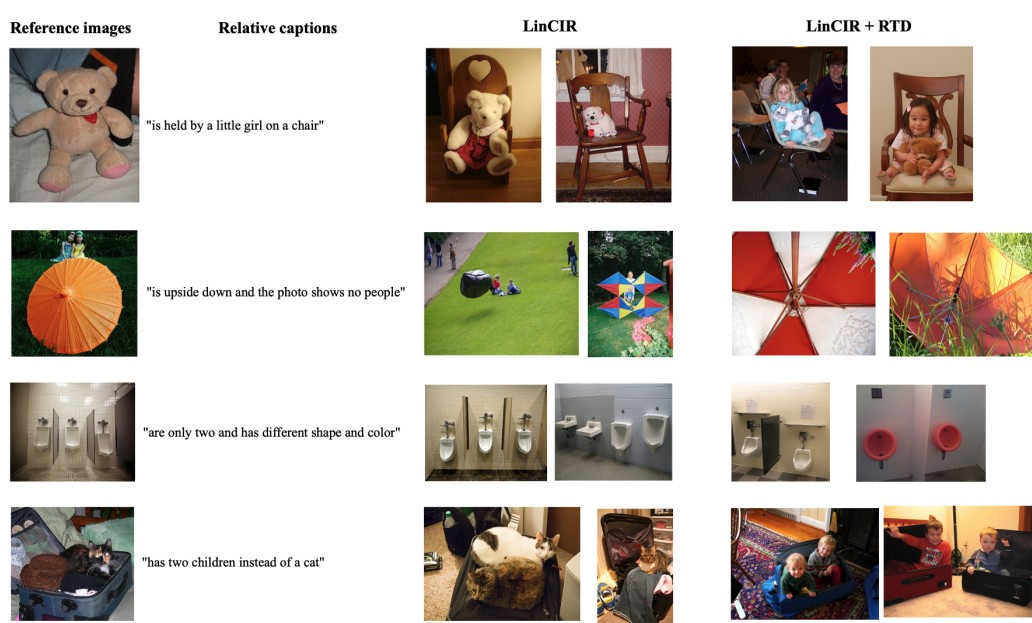

Figure D.1: Qualitative Results on CIRCO dataset

## E  DISCUSSION AND LIMITATIONS

We have primarily focused on evaluating the integrability of RTD with representative projection-based CIR methods (Saito et al., 2023; Baldrati et al., 2023; Gu et al., 2024). However, we have not yet explored or tested the extensibility of RTD to other CIR approaches that achieve strong performance, such as those utilizing human-annotated CIR triplets (supervised) (Baldrati et al., 2022b), synthetically generated CIR triplets (Ventura et al., 2024; Levy et al., 2024; Gu et al., 2023), or training-free methods (Karthik et al., 2023). Given the core motivation behind RTD, its adaptability to those CIR approaches that directly train fusion modules or backbones using CIR triplets may be limited. However, considering the strong performance and practical advantages—such as efficient

```
{
  "source_caption": "what do you do with automobile model for $60 k",
  "target_caption": "what do you do with model for $60 k",
  "relative_caption": "without automobile"
},
{
  "source_caption": "a collage of my latest artwork includes oil pastel and acrylic paintings",
  "target_caption": "a collage of my latest artwork includes water pastel and acrylic paintings",
  "relative_caption": "alter oil to match water"
},
{
  "source_caption": "baseball player hits a home run against sports team",
  "target_caption": "baseball customer hits a home run against sports team",
  "relative_caption": "player is removed and customer takes its place"
},
{
  "source_caption": "another wall at my home",
  "target_caption": "another bedroom at my home",
  "relative_caption": "bedroom is added in place of wall"
},
{
  "source_caption": "tennis player faces a tough schedule if she is to advance",
  "target_caption": "tennis player faces a tough routine if she is to advance",
  "relative_caption": "change schedule to routine"
},
```

Figure D.2: Example of rule-based triplet datasets

```
{
  "source_caption": "Christopher Nolan got advice from Steven Spielberg before making",
  "target_caption": "Steven Spielberg got advice from Walter Mitty before making",
  "relative_caption": "get advice from Walter Mitty"
},
{
  "source_caption": "by Koh Chip Whye — Black & White Buildings & Architecture",
  "target_caption": "by Koh Chip Whye — Colorful Buildings & Architecture",
  "relative_caption": "make the buildings more colorful"
},
{
  "source_caption": "The Most Hyperrealistic Images Of Beautiful Bathing Women With Their Heads Underwater",
  "target_caption": "The Most Hyperrealistic Images Of Beautiful Bathing Women With Their Heads Underwater and Octopus Arms",
  "relative_caption": "make the women have octopus arms"
},
{
  "source_caption": "Mountains above clouds — p312m1472749 by Mikael Svensson",
  "target_caption": "Mountains on Mars — p312m1472749 by Mikael Svensson",
  "relative_caption": "Put the mountains on Mars"
},
{
  "source_caption": "Le bouquiniste Paris —60x60",
  "target_caption": "The New York City book store —60x60",
  "relative_caption": "Instead of Paris, make it New York."
},
```

Figure D.3: Example of LLM-based triplet datasets

training and inference—offered by projection-based CIR methods compared to other variants, we believe that integrating RTD with them remains a valuable direction in the CIR domain.

## F SOCIETAL IMPACTS

Although our paper demonstrates promising outcomes in the ZS-CIR task, further examination of the data and the model is essential prior to practical deployment. Since our method focuses mainly

Table D.1: The full 50 keyword converting templates

| | |
|---|---|
| "replace ${source} with ${target}" | "substitute ${target} for ${source}" |
| "apply ${target}" | "${source} is removed and ${target} takes its place" |
| "convert ${source} to ${target}" | "modify ${source} to become ${target}" |
| "replace ${source} with ${target}" | "customize ${source} to become ${target}" |
| "update ${source} to ${target}" | "change ${source} to match ${target}" |
| "substitute ${target} for ${source}" | "${target} is introduced after ${source} is removed" |
| "alter ${source} to match ${target}" | "${target} is added in place of ${source}" |
| "upgrade ${source} to ${target}" | "${target} is introduced as the new option after" |
| "amend ${source} to fit ${target}" | "${source} is removed and ${target} is added" |
| "opt for ${target}" | "${source} is removed and ${target} is introduced" |
| "${source} is removed" | "${target} is added as a replacement for ${source}" |
| "add ${target}" | "${target} is the new option available" |
| "if it is ${target}" | "${target} is added after ${source} is removed" |
| "${target} is the updated option" | "${target} is introduced after ${source} is retired" |
| "${target} is the updated choice" | "tweak ${source} to become ${target}" |
| "${source} is replaced with ${target}" | "has no ${source}" |
| "change ${source} to ${target}" | "alter ${source} to ${target}" |
| "swap ${source} for ${target}" | "redesign ${source} as ${target}" |
| "turn ${source} into ${target}" | "adapt ${source} to fit ${target}" |
| "choose ${target} instead of ${source}" | "${target} is the new choice" |
| "${target} is the new selection" | "exchange ${source} with ${target}" |
| "transform ${source} into ${target}" | "show no ${source}" |
| "no ${source}" | "remove ${source}" |
| "delete ${source}" | "not a ${source}" |
| "with no ${source}" | "without ${source}" |

on optimization for accuracy, unwanted social implications can occur. For example, real-world images from databases and user-generated text may inadvertently cause harmful cases.

