# OpenReview forum: "Reducing Task Discrepancy of text encoders for Zero-Shot Composed Image Retrieval"
_ICLR.cc/2025/Conference — Submitted to ICLR 2025_

### Official Review · Reviewer_UMZF · 2024-10-31

**Soundness:** 3
**Presentation:** 3
**Contribution:** 2
**Rating:** 3
**Confidence:** 5

**Summary:**

This paper defines the task discrepancy between the CIR task and the CLIP model's pre-training task, and based on this, proposes an improved method that solely updates the text encoder using inexpensive text triplets, which consist of reference and target texts instead of images. Furthermore, integrating RTD into existing projection-based zero-shot composed image retrieval methods enhances performance across various datasets and backbones.

**Strengths:**

1. The idea presented in this paper is clear, and the writing structure is fluid;
2. The illustrations are visually appealing and facilitate the reader's understanding of the proposed method;
3. The proposed method provides benefits to three existing methods.

**Weaknesses:**

1. Firstly, this paper **lacks a thorough review of recent work**, as several representative studies have not been cited or analyzed, which diminishes its timeliness. For instance, in the zero-shot domain, there are three works [1-3] that were not compared or analyzed; regarding the task discrepancy of the CLIP model, it seems that an analysis has already been conducted in [4], but a comparative analysis was also lacking;
2. Based on the aforementioned issues, the proposed method is **insufficiently validated experimentally**. Although it has been tested on three methods, it should be demonstrated on a broader range of ZCIR methods, particularly more recently proposed methods with better performance, to further establish that the proposed method is indispensable and that other methods have not considered these aspects.

[1] Fine-grained Textual Inversion Network for Zero-Shot Composed Image Retrieval, SIGIR 2024.
[2] Image2Sentence based Asymmetric zero-shot composed image retrieval, ICLR 2024.
[3] LDRE: LLM-based Divergent Reasoning and Ensemble for Zero-Shot Composed Image Retrieval, SIGIR 2024.
[4] FashionERN: Enhance-and-Refine Network for Composed Fashion Image Retrieval, AAAI 2024.

**Questions:**

My questions primarily revolve around the paper's timeliness and experimental aspects; please refer to the Weaknesses section. Additionally, I am curious about the **quality of the captions** in the cheap text triplets.

**Details Of Ethics Concerns:**

No ethics concerns.

---

> ### Comment · Reviewer_UMZF · 2024-12-02
>
> Given that the authors have not addressed my queries regarding the novelty and experiments of the paper, which have also been questioned by other reviewers, I maintain my previous rating and level of confidence.

---

### Official Review · Reviewer_JNTt · 2024-10-31

**Soundness:** 3
**Presentation:** 2
**Contribution:** 2
**Rating:** 3
**Confidence:** 5

**Summary:**

This paper introduces RTD to fine-tune the CLIP language encoder for ZS-CIR models. RTD attempts to utilize rule-based templates or LLM-based triplets to address the task discrepancy inherent in the CLIP language encoder for ZS-CIR. The method also incorporates a refined batch sampling strategy for hard negatives and a sophisticated concatenation scheme to enhance the fine-tuning process. Moreover, the paper demonstrates that fine-tuning a part of the text encoder results comparable to full encoder adjustments. RTD enhances performance across various ZS-CIR backbones when integrated into SOTA projection-based ZS-CIR methods.

**Strengths:**

1. Generating text triplets without source images for fine-tuning is an interesting idea.

2. The results in only updating interleave 3 FCs, which could achieve a comparable performance, are supervised .

3. The various ablation studies show the effectiveness of RTD.

**Weaknesses:**

1. The setting of this paper is inconsistent with standard ZS-CIR tasks [1,2,3,4,5,6,7], which is unfair. This inconsistency setting may cause potential data leakage. The setting that leverages the synthetically generated CIR triplets (or text triplets) for fine-tuning the CLIP language encoder aims to make it more suitable for CIR, but it may not directly address the challenge of task Discrepancy. Moreover, including the synthetically generated triplets in fine-tuning may cause the CLIP language encoder fitting bias of the CIR data, leading to potential data leakage. The possible evidence might be seen in Table B.5, which only updated interleave 3 FCs could achieve a comparable performance.

2. The novelty is limited. The primary motivation involves fine-tuning the CLIP language encoder to make it more suitable for the input of the ZS-CIR model's pseudo-token, which is a straightforward way. It may limit its inspirational value for future works.

3. The technology contribution is limited. The text triplets are from existing methods, and the method for generating rule-based text triplets is similar to Compodiff. The concept of fine-tuning the CLIP model with pseudo-triplets to reduce task discrepancy in ZS-CIR derives from [8]; this paper seems only to propose a method to avoid the break of CLIP alignment knowledge, which is similar to the Momentum Distillation [9]. Moreover, RID needs to be re-trained for different ZS-CIR methods. Therefore, I would like to question whether RTD might be a trick rather than a method. It may not address key challenges for composed image retrieval tasks or improve the structure of existing ZS-CIR models. Instead, the authors might simply find that the ZS-CIR model's pseudo-token is not well understood by the CLIP language encoder, so they directly fine-tuned it.

4. The efficiency analysis is unfair in lines 264-280. The proposed method is a "step-by-step" approach. It should first train the original ZS-CIR method and then use RTD to fine-tune the CLIP language encoder for each ZS-CIR method based on the different pseudo-token. However, in Lines 277-278, the authors compare the time cost between the original ZS-CIR model's training process and RID's fine-tuning process, which is unfair. Furthermore, I question the training cost, which is huge, as compared to pic2word(ViT-L/14), 4 V100 is needed, and the requirement of RTD increases to 8 A800. Moreover, for the larger backbone (e.g., ViT-G-14), the cost of fine-tuning the CLIP language encoder may have significant increments, which the paper does not report. This point significantly influences the contribution of RTD.

5. The comparison of the main results is incomplete. For example, the SoTA of the CIRCO dataset is CIReVL(ViT-G-14) [6] in Table 5, which this paper does not compare. Additionally, the comparison of the main results is messy and confusing.

Overall, due to the unfair setting with potential data leakage, limited novelty and technology contribution, and unfair competition in efficiency, I give a "Reject" recommendation. I will consider raising my score if the authors address my concerns.

Reference

[1] Kuniaki Saito, Kihyuk Sohn, Xiang Zhang, Chun-Liang Li, Chen-Yu Lee, Kate Saenko, and Tomas Pfister. Pic2word: Mapping pictures to words for zero-shot composed image retrieval. In CVPR, 2023.

[2] Geonmo Gu, Sanghyuk Chun, Wonjae Kim, Yoohoon Kang, and Sangdoo Yun. Language-only efficient training of zero-shot composed image retrieval. In CVPR, 2024.

[3] Suo Y, Ma F, Zhu L, et al. Knowledge-enhanced dual-stream zero-shot composed image retrieval[C]//Proceedings of the IEEE/CVF Conference on Computer Vision and Pattern Recognition. 2024: 26951-26962.

[4] Tang Y, Yu J, Gai K, et al. Context-I2W: Mapping Images to Context-dependent Words for Accurate Zero-Shot Composed Image Retrieval[C]//Proceedings of the AAAI Conference on Artificial Intelligence. 2024, 38(6): 5180-5188.

[5] Du Y, Wang M, Zhou W, et al. Image2Sentence based Asymmetrical Zero-shot Composed Image Retrieval[J]. ICLR 2024.

[6] Karthik S, Roth K, Mancini M, et al. Vision-by-language for training-free compositional image retrieval[J]. ICLR 2024.

[7] Alberto Baldrati, Lorenzo Agnolucci, Marco Bertini, and Alberto Del Bimbo. Zero-shot composed image retrieval with textual inversion. In ICCV, 2023.

[8] Junyang Chen and Hanjiang Lai. Pretrain like you inference: Masked tuning improves zero-shot composed image retrieval. arXiv preprint arXiv:2311.07622, 2023.

[9] Li J, Selvaraju R, Gotmare A, et al. Align before fuse: Vision and language representation learning with momentum distillation[J]. Advances in neural information processing systems, 2021, 34: 9694-9705.

**Questions:**

1. What is the fine-tuning cost (e.g., training parameter size, GPU cost)  for lager backbones (i.e., ViT-G/14)?

1. What is the fine-tuning time for different CLIP backbones?

2. Why do you not compare the best result of CIReVL (i.e., ViT-G-14)? It is suggested that you compare other methods in your main results.

---

> ### Comment · Reviewer_JNTt · 2024-12-02
>
> Unfortunately, the authors have not provided any responses to my previous concerns regarding the setting, novelty, and unfair efficiency analysis of the paper. Other reviewers also raised similar concerns. Given the lack of any reply or clarification, I maintain my previous rating and level of confidence.

---

### Official Review · Reviewer_URQq · 2024-11-01

**Soundness:** 3
**Presentation:** 2
**Contribution:** 2
**Rating:** 3
**Confidence:** 5

**Summary:**

This paper addresses the paradigm differences between the pre-training and inference stages of the existing Zero-Shot Composed Image Retrieval task. A target-anchored text contrastive learning approach is proposed, which replaces images in the triplet with their captions for model training. The effectiveness of the proposed method is validated across multiple benchmarks.

**Strengths:**

1. The proposed method in this paper can serve as a plug-in to enhance the performance of existing zero-shot methods.
2. The illustrations are clear and detailed, facilitating understanding.

**Weaknesses:**

1. The motivation presented is not novel enough. Previous work [a] has already addressed the differences between the triplet data used by the CLIP encoder and the pre-trained text-image pairs in CIR tasks. Paper [b] also focuses on methods that only train the text encoder. The authors should further elaborate on the originality of their approach.
2. The authors need to explain and justify why training only the text encoder is superior to training the image encoder. For instance, they should clarify whether training just the text encoder can achieve fine-grained capabilities, as shown in paper [c].
3. Although the proposed method demonstrates some improvements, it still falls short compared to state-of-the-art methods [c][d]. Moreover, ablation studies indicate that the enhancement of RB in the CIRR scenario is quite limited.

[a] Chen. et al. FashionERN: Enhance-and-Refine Network for Composed Fashion Image Retrieval. AAAI 2024.
[b] Gu. Et al. Language-only efficient training of zero-shot composed image retrieval. CVPR 2024.
[c] Lin. et al. Fine-grained Textual Inversion Network for Zero-Shot Composed Image Retrieval. SIGIR 2024.
[d] Du. et al. Image2Sentence based Asymmetrical Zero-shot Composed Image Retrieval. ICLR 2024.

**Questions:**

Please refer to the Weaknesses section and address my concerns regarding the methodological innovation, and performance of the proposed method.

---

### Official Review · Reviewer_4GNM · 2024-11-02

**Soundness:** 2
**Presentation:** 2
**Contribution:** 2
**Rating:** 3
**Confidence:** 4

**Summary:**

This paper proposes a Reducing Task Discrepancy method for the zero-shot composed image retrieval task. The proposed method designs target-anchored text contrastive learning to update the text encoder using cheap text triplets consisting of reference and target texts instead of images.

**Strengths:**

1. Comparison experiments demonstrate the effectiveness of the proposed method.

**Weaknesses:**

1. The motivation and the novelty of this paper requires further clarification. The idea of generating pseudo triplets is common for the zero-shot composed image retrieval task, and whether the task discrepancy is reduced need more analyses to validate.
2. More visualiztion results should be provided to further validate the effectiveness of the proposed method, such as top-10 retrieval results, and the correct and wrong results should be highlighted respectively.
3. The best and second-best results should be highlighted respectively for better comprehension.

**Questions:**

1. The constructed triplets only contain text modality. Why not construct triplets with modified texts and target images to better imitate the scenario of the ZS-CIR task?
2. The key challenge of reducing task discrepancy is concerning the text-image modality gap and the (text+image)-image modality gap. More statistical or theoretical analyses with respect to the mixture modality of modified texts and reference images should be conducted to validate the reduction on task discrepancy.

---

### Official Review · Reviewer_vbiZ · 2024-11-02

**Soundness:** 3
**Presentation:** 3
**Contribution:** 2
**Rating:** 5
**Confidence:** 4

**Summary:**

This paper studies Zero-Shot Composed Image Retrieval (ZS-CIR), which aims to retrieve a target image based on a reference image and conditioning text, enabling controllable searches. The authors construct a purely textual dataset to further pretrain CLIP, using Reducing Task Discrepancy of text encoders for Composed Image Retrieval (RTD). The trained CLIP is effective and can be integrated into many ZS-CIR methods. Experimental results also show the effectiveness of the proposed method.

**Strengths:**

1. The proposed method is rational, efficient, and flexible to be combined with many ZS-CIR methods.
2. Extensive experiments analyze the proposed method and show its effectiveness.
3. The writing is good.

**Weaknesses:**

1. The evaluation of the ZS-CIR methodology presented in this paper may not align with conventional standards. Existing ZS-CIR methods leverage a frozen CLIP model. In contrast, this approach fine-tunes the CLIP model, thereby deviating from the standard configurations of published ZS-CIR models. This adjustment does not adhere to a true "zero-shot" paradigm within the ZS-CIR framework.
2. I raise concerns about the fairness of this methodology, as it may introduce the risk of data leakage during the training phase.
3. Fine-tuning only the text encoder does not adequately address the issue of task discrepancy in ZS-CIR. This problem primarily stems from the differences between training and inference phases [1]. The authors have not altered the training strategy of the baseline model; they have simply fine-tuned the text encoder while keeping other parameters constant.

References:
[1] Karthik S, Roth K, Mancini M, et al. Vision-by-language for training-free compositional image retrieval[J]. ICLR 2024.

**Questions:**

Please respond to Weaknesses.

---

### Meta-Review · Area_Chair_PJHE · 2024-12-18

**Metareview:**

The paper tackles Zero-Shot Composed Image Retrieval by refining the CLIP model with a purely textual dataset to better align the pre-training and inference stages. By introducing RTD and using text triplets for fine-tuning, the method enhances performance across multiple benchmarks and existing ZS-CIR models.



The reviewers appreciate the proposed method is adaptable, efficient and can be smoothly integrated iwth many ZS-CIR methods and enhance their performance. The authors also acknowledge that the experiments validate the effiectiveness of method. However, there are several concerns and questions are raised. 1. Limited novelty and contribution. Some reviewers think the methodology part lack originality and significant technical contributions. 2. Deviation from standard methods, raise the concerns about data leakage and fairness. 3. Experimental evaluation concerns. The evaluation seems incomplete, requiring more comprehensive comparison. Given all these, I’m recommending a reject.

**Additional Comments On Reviewer Discussion:**

No rebuttal provided.

---

### Decision · Program_Chairs · 2025-01-22

Reject